# Cyclic AMP is a critical mediator of intrinsic drug resistance and fatty acid metabolism in *M. tuberculosis*

Andrew I Wong[1], Tiago Beites[2], Kyle A Planck[2,3], Rachael A Fieweger[4], Kathryn A Eckartt[1], Shuqi Li[1], Nicholas C Poulton[1], Brian C VanderVen[4], Kyu Y Rhee[2,3], Dirk Schnappinger[2], Sabine Ehrt[2], Jeremy Rock[1]*

[1]Laboratory of Host-Pathogen Biology, The Rockefeller University, New York, United States; [2]Department of Microbiology and Immunology, Weill Cornell Medicine, New York, United States; [3]Division of Infectious Diseases, Department of Medicine, Weill Cornell Medicine, New York, United States; [4]Department of Microbiology and Immunology, College of Veterinary Medicine, Cornell University, Ithaca, United States

*For correspondence:
rock@rockefeller.edu

Competing interest: The authors declare that no competing interests exist.

**ABSTRACT** Cyclic AMP (cAMP) is a ubiquitous second messenger that transduces signals from cellular receptors to downstream effectors. *Mycobacterium tuberculosis* (Mtb), the etiological agent of tuberculosis, devotes a considerable amount of coding capacity to produce, sense, and degrade cAMP. Despite this fact, our understanding of how cAMP regulates Mtb physiology remains limited. Here, we took a genetic approach to investigate the function of the sole essential adenylate cyclase in Mtb H37Rv, Rv3645. We found that a lack of *rv3645* resulted in increased sensitivity to numerous antibiotics by a mechanism independent of substantial increases in envelope permeability. We made the unexpected observation that *rv3645* is conditionally essential for Mtb growth only in the presence of long-chain fatty acids, a host-relevant carbon source. A suppressor screen further identified mutations in the atypical cAMP phosphodiesterase *rv1339* that suppress both fatty acid and drug sensitivity phenotypes in strains lacking *rv3645*. Using mass spectrometry, we found that Rv3645 is the dominant source of cAMP under standard laboratory growth conditions, that cAMP production is the essential function of Rv3645 in the presence of long-chain fatty acids, and that reduced cAMP levels result in increased long-chain fatty acid uptake and metabolism and increased antibiotic susceptibility. Our work defines *rv3645* and cAMP as central mediators of intrinsic multidrug resistance and fatty acid metabolism in Mtb and highlights the potential utility of small molecule modulators of cAMP signaling.

## Editor's evaluation

Bacteria living in stressful and fluctuating environments need to respond to changing conditions, and many species, including *Mycobacterium tuberculosis*, the causative agent of tuberculosis, use cyclic AMP (cAMP) as a secondary messenger to sense and respond to specific stimuli. What distinguishes *M. tuberculosis*, is that its genome encodes at least 15 adenylate cyclases, enzymes that synthesize cAMP from ATP. Using state-of-the-art methods in this important study, the authors characterized one specific adenylate cyclase, Rv3645, and convincingly demonstrate that it is the most significant contributor to cAMP levels, in addition to mediating fatty acid metabolism and antibiotic resistance. This manuscript will be of broad interest to readers in the field of tuberculosis drug discovery and bacterial metabolism.

## Introduction

*Mycobacterium tuberculosis* (Mtb) has evolved complex signaling and regulatory networks to sense and adapt to the diverse niches through which it transits during infection (*Johnson and McDonough, 2018*; *Parish, 2014*; *Richard-Greenblatt and Av-Gay, 2017*). The physiology that enables Mtb to adapt to and persist in the host can also decrease the effectiveness of antibiotics (*Bellerose et al., 2020*; *Larrouy-Maumus et al., 2016*). For example, hypoxia reduces Mtb respiratory capacity and activates the DosRST two-component system (*Park et al., 2003*). DosR induces a 48 gene regulon which ultimately slows growth, promoting survival under hypoxic conditions and tolerance to antibiotics that are more active against rapidly replicating bacteria (*Galagan et al., 2013*). In a second example, nutrient starvation results in the dephosphorylation of CwlM, a substrate of the eukaryotic-like protein serine/threonine kinase PknB (*Boutte et al., 2016*). Dephosphorylation reduces CwlM interaction with and activation of the peptidoglycan biosynthetic enzyme MurA, thereby reducing cell wall metabolism and promoting tolerance to starvation and antibiotics. Thus, the signaling and regulatory networks that facilitate Mtb survival under diverse physiologic conditions can also secondarily reduce the effectiveness of antibiotics. While several examples have been described, it is clear that numerous poorly understood signaling mechanisms exist that promote Mtb survival while reducing the effectiveness of antibiotic therapy (*Bellerose et al., 2020*).

In addition to two-component systems and serine/threonine kinases, one of the most ubiquitous signal transduction modalities in Mtb is the adenylate cyclases. Adenylate cyclases sense extracellular or intracellular signals, either directly or indirectly, and transduce this signal into a cellular response by converting ATP into the small molecule second messenger 3',5'-cyclic-AMP (cAMP) and pyrophosphate (*Johnson and McDonough, 2018*). cAMP then binds to and alters the function of effector proteins like the transcription factor CRP (*Stapleton et al., 2010*), the protein lysine acetyltransferase Mt-Pat (*Nambi et al., 2013*), and numerous other potential cAMP-binding proteins (*Johnson and McDonough, 2018*) to modify Mtb gene expression or gene product activity. Whereas the model bacteria *E. coli* encodes only one adenylate cyclase, the adenylate cyclase gene family has undergone a remarkable expansion in mycobacteria. *M. avium*, *M. marinum*, and Mtb encode 12, 31, and at least 15 predicted adenylate cyclases (*Shenoy and Visweswariah, 2006*), respectively. The existence of so many adenylate cyclases in the Mtb genome presumably allows Mtb to integrate diverse signals with downstream cellular responses by using cAMP as a second messenger. Mtb adenylate cyclases can be activated by a variety of stimuli, including pH (*Tews et al., 2005*), bicarbonate (*Cann et al., 2003*), and fatty acids (*Abdel Motaal et al., 2006*), with the resulting increase in cAMP levels modulating both bacterial and host physiology (*Agarwal et al., 2009*). This expansion of the adenylate cyclase gene family is mirrored by an expansion of predicted cAMP phosphodiesterases, effectors, and binding proteins (*Figure 1—figure supplement 1A*) – nearly 1% of the Mtb genome is predicted to produce, degrade, or interact with cAMP. However, despite the clear importance of cAMP signaling in mycobacteria, our knowledge of how adenylate cyclases and cAMP regulate Mtb physiology remains largely undefined.

To address this gap, we undertook a genetic study of the sole in vitro essential adenylate cyclase in Mtb H37Rv, *rv3645*. We found that a lack of *rv3645* resulted in increased sensitivity to numerous antibiotics by a mechanism independent of substantial increases in envelope permeability. We further found that *rv3645* was conditionally essential for Mtb growth in the presence of long-chain fatty acids, a host-relevant carbon source, and identified mutations in the atypical cAMP phosphodiesterase *rv1339* that suppress this fatty-acid-dependent essentiality. Quantifying cAMP levels revealed that Rv3645 is the dominant source of cAMP under standard laboratory growth conditions. Bacterial cAMP levels were not altered by the presence of long-chain fatty acids nor the presence of some antibiotics to which *rv3645* mutants were more sensitive, suggesting that Rv3645 does not sense the presence of long-chain fatty acids or antibiotics. Rather, cAMP produced by Rv3645, presumably in response to an alternative signal, puts the bacilli in a physiologic state capable of surviving the stresses of long-chain fatty acids and antibiotics. Together, our work defines Rv3645 and the ubiquitous second messenger cAMP as central mediators of intrinsic multidrug resistance and fatty acid metabolism in Mtb.

# Results

## The essential adenylate cyclase Rv3645 contributes to intrinsic drug resistance in Mtb H37Rv

To identify genes and pathways that influence drug efficacy in Mtb, we previously screened a genome-wide CRISPRi library in Mtb strain H37Rv against a panel of diverse antibiotics (*Li et al., 2022*). This approach identified hundreds of Mtb genes whose inhibition altered bacterial fitness in the presence of partially inhibitory drug concentrations, including genes encoding the direct drug target and non-target hit genes. Amongst the non-target hit genes, we found that depletion of the predicted essential adenylate cyclase, *rv3645* (*Figure 1A*), sensitized Mtb to numerous antibiotics with unrelated mechanisms of action (*Figure 1B*). Mtb H37Rv encodes 15 putative adenylate cyclases that are expressed at various levels in standard laboratory culture conditions (*Figure 1—figure supplement 1A,B*). Interestingly, *rv3645* was the only adenylate cyclase whose knockdown resulted in increased drug sensitivity (*Figure 1—figure supplement 1A*), demonstrating a lack of substantial functional redundancy for adenylate cyclases under these culture conditions. Given the predicted essentiality of *rv3645* and the magnitude by which silencing of this gene sensitized Mtb to various antibiotics, we next sought to better characterize the role of this gene in Mtb physiology.

We first sought to validate the screen results. To do this, we cloned a single-guide RNA (sgRNA) targeting *rv3645* into an inducible CRISPRi plasmid that allows for targeted *rv3645* knockdown in the presence of anhydrotetracycline (ATc) (*Rock et al., 2017*). We also cloned complementation constructs expressing CRISPRi-sensitive or CRISPRi-resistant *rv3645* alleles (*Wong and Rock, 2021*). These *rv3645* complementation alleles differ only by silent mutations within the *rv3645* ORF that abrogate CRISPRi targeting in the resistant allele but do not affect the wild-type protein sequence. Consistent with prior screens (*Bosch et al., 2021*; *DeJesus et al., 2017*), knockdown of *rv3645* prevented growth on 7H10-OADC agar plates (*Figure 1C*). This growth defect was complemented by expressing a CRISPRi-resistant but not a CRISPRi-sensitive *rv3645* allele, demonstrating that growth inhibition was indeed a result of silencing *rv3645*. To validate the results of the CRISPRi chemical-genetic screen, we measured the minimum inhibitory concentrations (MICs) of a panel of antibiotics against the *rv3645* CRISPRi strains. Consistent with the CRISPRi screening results, *rv3645* knockdown sensitized Mtb to vancomycin, rifampicin, clarithromycin, bedaquiline, and meropenem but not other drugs (*Figure 1D–G*; *Figure 1—figure supplement 1C–H*). These results validate that the sole essential adenylate cyclase Rv3645 contributes to the intrinsic resistance of Mtb H37Rv to various antibiotics.

## Increased drug sensitivity in *rv3645* knockdown strains is not due to large increases in envelope permeability

The mycobacterial cell envelope serves as a permeability barrier that restricts access to antibiotics to their intracellular or periplasmic targets (*Batt et al., 2020*). The similarities between the chemical-genetic signatures of *rv3645* and envelope biosynthetic genes (*Figure 2A*; *Li et al., 2022*) suggested that *rv3645* may contribute to intrinsic drug resistance by promoting envelope integrity in Mtb. To test this hypothesis, we used a fluorescent, BODIPY-conjugated analog of vancomycin (BODIPY-VAN) to monitor drug uptake. Vancomycin is a large, polar antibiotic for which disruption of the Mtb envelope is known to increase drug uptake and increase drug sensitivity (*Li et al., 2022*). Surprisingly, despite the dramatic sensitization of *rv3645* knockdown strains to vancomycin (*Figure 1D*), *rv3645* knockdown strains showed only a modest increase in BODIPY-VAN uptake (*Figure 2B*) as compared to the positive control gene *mtrA*, encoding a two-component response regulator important for proper envelope biogenesis (*Li et al., 2022*). To ensure the discrepancy between *rv3645* MIC assays and drug uptake measurements was not due to the presence of the BODIPY conjugate (~274 Daltons), we first confirmed that BODIPY-VAN retained similar antimicrobial activity to vancomycin (*Figure 2—figure supplement 1A*). To ensure that the BODIPY-VAN uptake assay was not confounded by differential cell death and lysis (cells which presumably would not stain with BODIPY-VAN), we confirmed that no loss in colony forming units (CFU) occurred when treating this panel of Mtb strains with vancomycin for the same time scale as the uptake assay (*Figure 2—figure supplement 1B*). To further test the uptake hypothesis, we next monitored drug uptake directly by mass spectrometry. Consistent with the BODIPY-VAN results, *rv3645* knockdown strains did not show elevated levels of vancomycin uptake (*Figure 2C*). Finally, we monitored

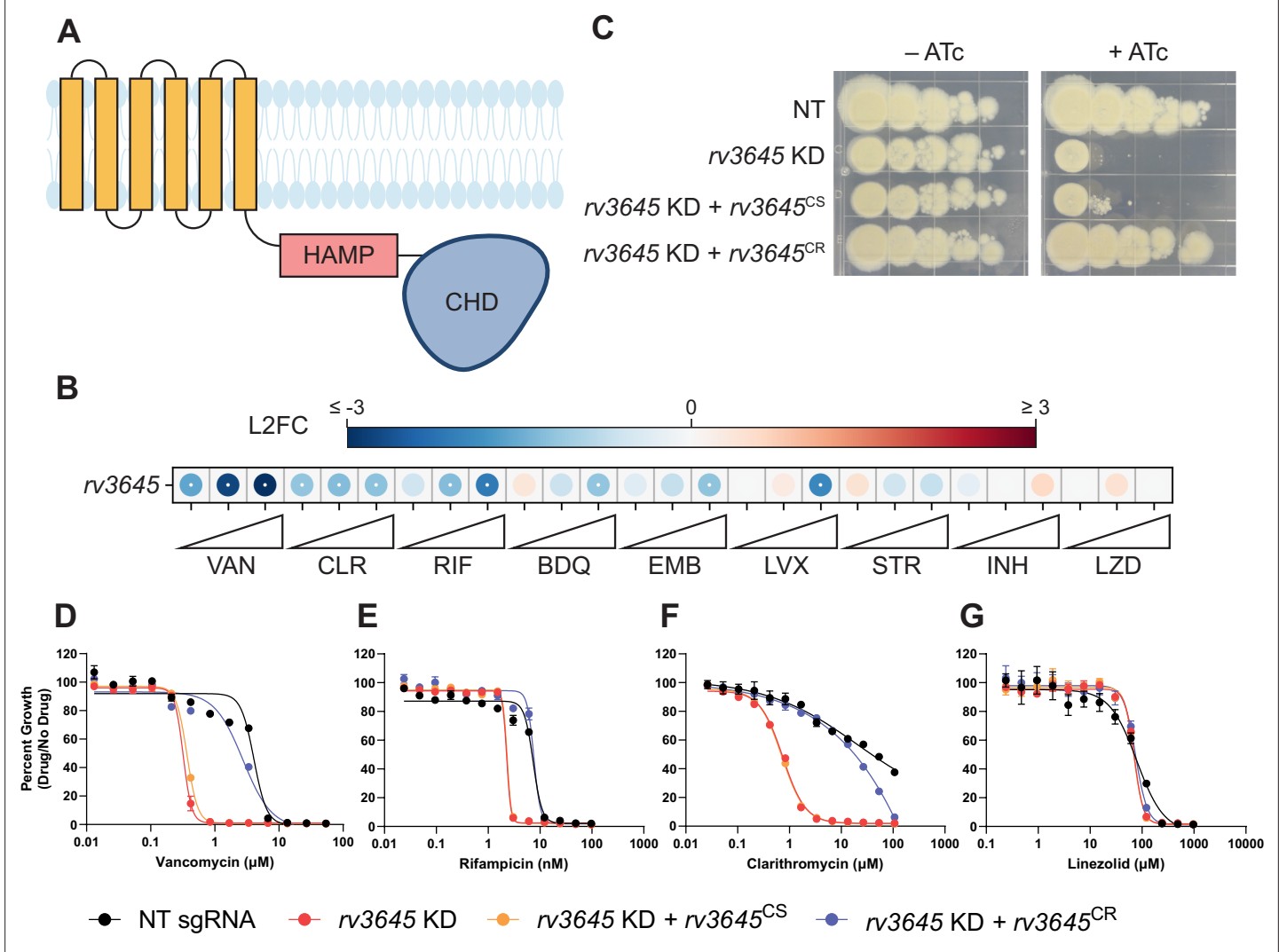

**Figure 1.** The adenylate cyclase *rv3645* is critical for intrinsic multidrug resistance in *Mycobacterium tuberculosis (Mtb)*. (**A**) Predicted domain organization of Rv3645. N-terminal transmembrane helices, HAMP domain, and C-terminal cytosolic adenylate cyclase domain (CHD) are shown. HAMP = Histidine kinases, Adenylate cyclases, Methyl-accepting proteins, and Phosphatases; CHD = Cyclase Homology Domain. (**B**) Feature-expression heatmap of *rv3645* from a 5 day CRISPRi library pre-depletion screen. The color of each circle represents the gene-level log2 fold change (L2FC); a white dot represents a false discovery rate (FDR) <0.01 and a |L2FC|>1. VAN = vancomycin; CLR = clarithromycin; RIF = rifampicin; BDQ = bedaquiline; EMB = ethambutol; LVX = levofloxacin; STR = streptomycin; INH = isoniazid; LZD = linezolid. Each antibiotic was tested in triplicate at three sub-minimum inhibitory concentrations (sub-MIC90) listed in *Figure 1—source data 1*. (**C**) Growth of *rv3645* CRISPRi strains on 7H10-OADC agar. Columns represent 10-fold serial dilutions in cell number. NT = non-targeting sgRNA; KD = knockdown; CS = CRISPRi-sensitive; CR = CRISPRi-resistant. (**D–G**) Dose-response curves for (**D**) vancomycin, (**E**) rifampicin, (**F**) clarithromycin, and (**G**) linezolid were measured against *rv3645* CRISPRi strains. Data represent mean ± SEM for technical triplicates and are representative of at least two independent experiments.

The online version of this article includes the following source data and figure supplement(s) for figure 1:

**Source data 1.** Antibiotic concentrations (nanomolar) used for CRISPRi chemical-genetic interaction screens.

**Figure supplement 1.** *rv3645* contributes to multidrug intrinsic resistance in H37Rv Mycobacterium tuberculosis (Mtb).

envelope permeability with the reporter dyes ethidium bromide, calcein-AM, and BCECF-AM. As with the vancomycin uptake assays, *rv3645* knockdown strains were not hyperpermeable to these dyes, as compared to the positive control genes involved in envelope biosynthesis and integrity (*Figure 2D–F*). Taken together, these results demonstrate that depletion of *rv3645* does not result in large increases in Mtb envelope permeability. Thus, Rv3645-mediated intrinsic drug resistance must occur primarily by some other mechanism.

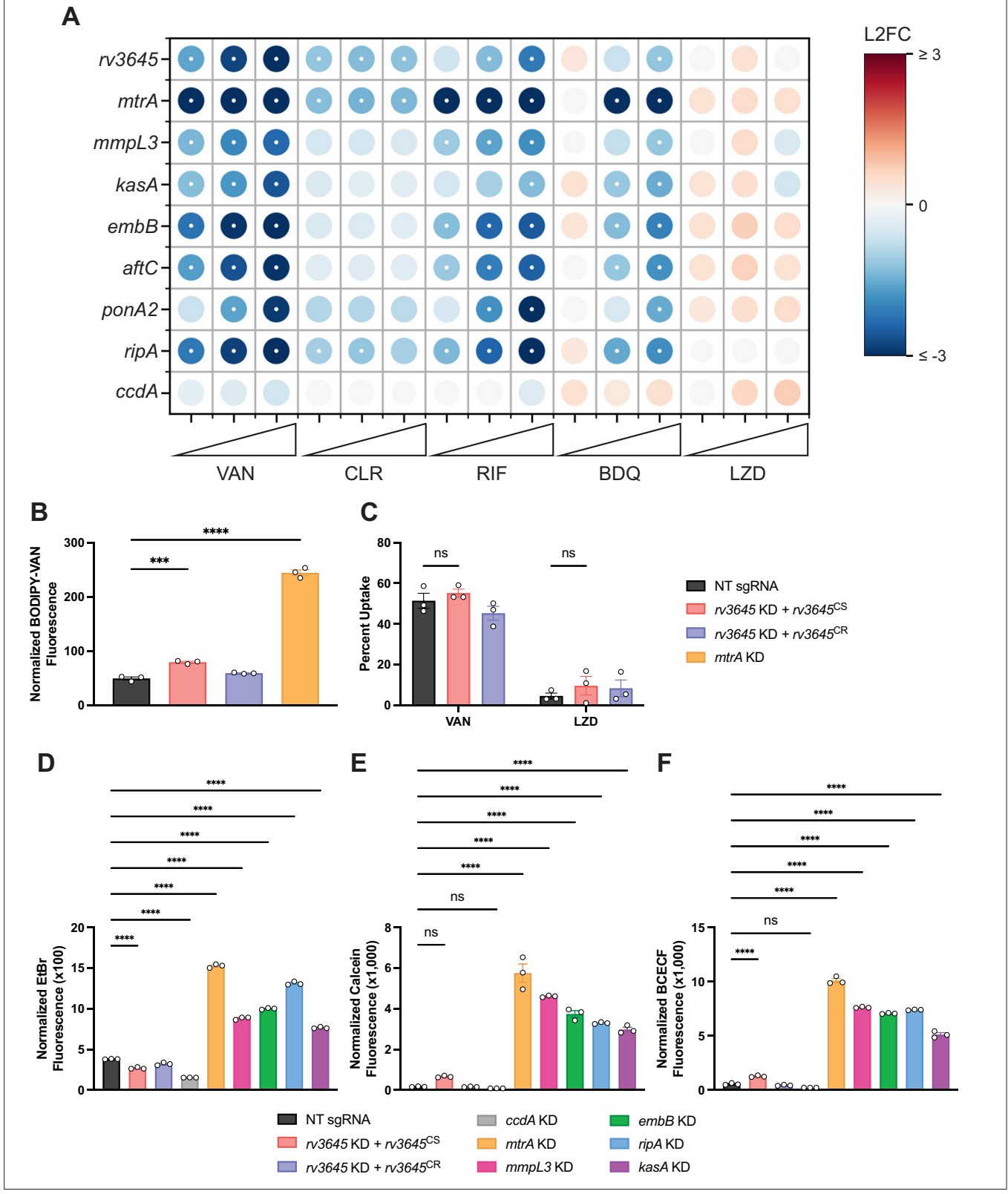

**Figure 2.** Increased drug sensitivity in *rv3645* knockdown strains is not due to large increases in envelope permeability. (**A**) Feature-expression heatmap of genes important for Mycobacterium tuberculosis (Mtb) cell envelope biosynthesis from the 5 day CRISPRi library pre-depletion screen for select drugs. *ccdA* is an in vitro essential non-hit control gene unrelated to envelope biosynthesis. The color of each circle represents the gene-level L2FC; a white dot represents a false discovery rate (FDR) of <0.01 and a |L2FC|>1. VAN = vancomycin; CLR = clarithromycin; RIF = rifampicin;

*Figure 2 continued*

BDQ = bedaquiline; LZD = linezolid. (**B**) BODIPY-Vancomycin uptake of the indicated strains. Data represent mean ± SEM for three replicates and are representative of two independent experiments. ***, p<0.001; ****, p<0.0001. Statistical significance was assessed by one-way ANOVA. (**C**) Quantification of vancomycin and linezolid uptake by mass spectrometry for the indicated strains. Data represent mean ± SEM for technical triplicates. ns = not significant. Statistical significance was assessed by two-way ANOVA (GraphPad Prism). Percent Uptake values are listed in *Figure 2—source data 1*. (**D–F**) Ethidium bromide (**D**), Calcein-AM (**E**), and BCECF-AM (**F**) uptake of the indicated strains. NT = non-targeting sgRNA; KD = knockdown; CS = CRISPRi-sensitive; CR = CRISPRi-resistant. Data represent mean ± SEM for three technical replicates and are representative of at least two independent experiments. ****, p<0.0001. Statistical significance was assessed by one-way ANOVA.

The online version of this article includes the following source data and figure supplement(s) for figure 2:

**Source data 1.** Antibiotic uptake in *rv3645* CRISPRi strains.

**Figure supplement 1.** Validation of the BODIPY-conjugated vancomycin uptake assay.

## *rv3645* essentiality and contribution to intrinsic drug resistance is conditional on the presence of long-chain fatty acids

*rv3645* is essential for Mtb H37Rv growth in standard laboratory media (7H10+OADC supplement: oleic acid, albumin, dextrose, and catalase; *Figure 1C*). Curiously, when the same growth medium was instead supplemented without fatty acid (ADC), *rv3645* knockdown no longer inhibited growth (*Figure 3—figure supplement 1A*), suggesting that *rv3645* is conditionally essential in the presence of oleic acid. Using fatty acid-free growth conditions, we were able to generate an *rv3645* deletion strain (Δ*rv3645*). Genetic identity was confirmed through whole genome sequencing. Plating of Δ*rv3645* in the presence or absence of oleic acid confirmed that *rv3645* is conditionally essential in the presence of this fatty acid (*Figure 3A*). To determine which other fatty acids may render *rv3645* essential, we measured MICs of fatty acids of increasing carbon chain lengths. Δ*rv3645* was uniquely sensitive to long-chain fatty acids palmitic acid (C16:0), oleic acid (C18:1), and arachidonic acid (C20:4); but not too short or medium-chain fatty acids or cholesterol (*Figure 3B–D*; *Figure 3—figure supplement 1B–G*). Notably, sensitivity was not observed towards the odd chain fatty acids propionic acid and valeric acid nor to cholesterol, ruling out propionate-derived toxicity as the source of the fatty acid-sensitive growth phenotype (*Eoh and Rhee, 2014*). The fact that the growth of Δ*rv3645* in the presence of long-chain fatty acids was not rescued by alternative carbon sources in the medium (glycerol and glucose) suggests that long-chain fatty acids are toxic to Δ*rv3645*, rather than Δ*rv3645* being unable to consume them. Consistent with this interpretation, Δ*rv3645* showed elevated uptake and metabolism of [1-$^{14}$C]-oleic acid (*Figure 3E and F*).

We next sought to determine if the contribution of *rv3645* to intrinsic drug resistance is also dependent on the presence of long-chain fatty acids. As with the growth defect, drug sensitivity associated with Δ*rv3645* was also dependent on the presence of long-chain fatty acids in the growth media (*Figure 3G–I*). These results demonstrate that *rv3645* essentiality and contribution to intrinsic drug resistance are both conditional on the presence of long-chain fatty acids.

## Loss-of-function of the atypical cAMP phosphodiesterase *rv1339* rescues fatty acid and drug sensitivity phenotypes of the Δ*rv3645* strain

Thus far, our results suggest that Rv3645 plays an important role in long-chain fatty acid metabolism in Mtb. To begin to interrogate how Rv3645 may contribute to lipid metabolism, we conducted a CRISPRi screen to identify suppressors of the fatty acid-dependent growth defect of Δ*rv3645*. An ATc-inducible CRISPRi library consisting of 96,700 sgRNAs targeting 4,054/4,125 of all Mtb genes was transformed into Δ*rv3645* (*Figure 4A*; *Bosch et al., 2021*). The resulting Δ*rv3645* CRISPRi library was then cultured on 7H10 agar in the presence of ATc and in the presence or absence of a toxic concentration of palmitic acid. As expected, growth was markedly reduced in the presence of palmitic acid. Colonies that grew in the presence of palmitic acid were expected to harbor sgRNAs that silence genes that contribute to fatty acid toxicity in Δ*rv3645*. Colonies were harvested from both culture conditions and their sgRNAs were PCR amplified, deep sequenced, and counted to compare sgRNA representation +/– palmitic acid. Hit genes were identified by MAGeCK (*Li et al., 2014*).

The suppressor screen identified nine enriched genes, eight of which encode structural or catalytic subunits of the Mce1 transporter (*Figure 4B*, *Figure 4—source data 1*). Mce1 has recently been

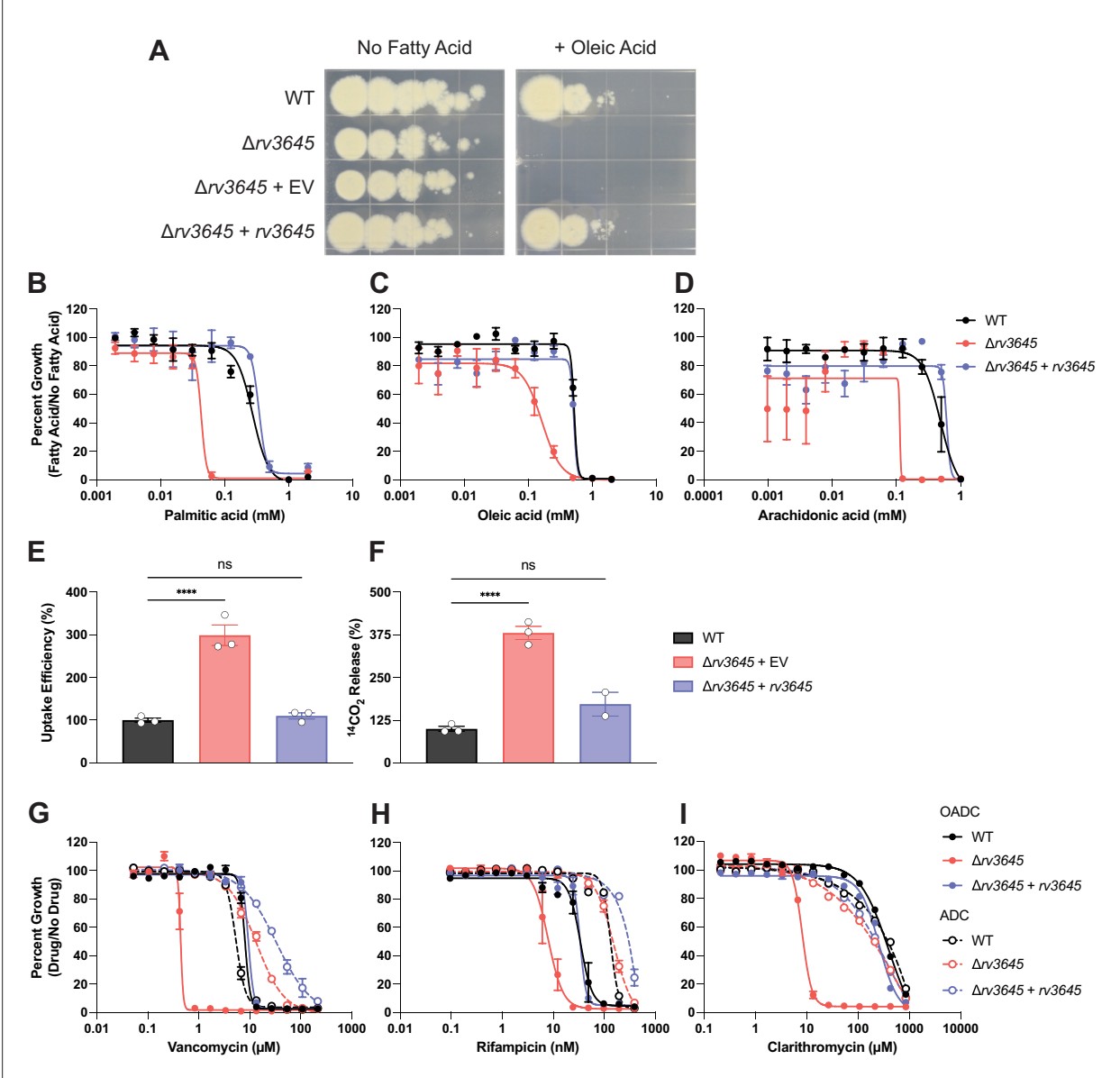

**Figure 3.** *rv3645* essentiality and contribution to intrinsic drug resistance is conditional on the presence of long-chain fatty acids. (**A**) Growth of Δ*rv3645* deletion strains on 7H10-ADC agar in the presence or absence of oleic acid. EV = empty vector. (**B–D**) Dose-response curves for fatty acids palmitic acid (**B**), oleic acid (**C**), and arachidonic acid (**D**). Data represent mean ± SEM for technical triplicates and are representative of at least three independent experiments. (**E**) Uptake of [1-$^{14}$C]-oleic acid in indicated strains. Uptake rates were calculated from the incorporated radioactive counts (*Figure 3— figure supplement 2*). Statistical significance was determined by one-way ANOVA. Data are representative of two experiments. (**F**) Catabolic release of $^{14}CO_2$ from [1-$^{14}$C]-oleic acid in the indicated strains. Data are normalized to cell number as estimated by $OD_{600}$, quantified relative to wild-type (WT), and represent means +/- SEM from technical triplicates, and are representative of two experiments. OE = over-expression, ns = not significant; ****, $p<0.0001$. Statistical significance was determined by one-way ANOVA. (**G–I**) Dose-response curves for vancomycin (**G**), rifampicin (**H**), and clarithromycin (**I**) of the indicated strains grown in 7H9 with (OADC) or without (ADC) oleic acid. Data represent mean ± SEM for technical triplicates and are representative of at least two independent experiments.

The online version of this article includes the following source data and figure supplement(s) for figure 3:

**Figure supplement 1.** Δ*rv3645* does not sensitize *Mycobacterium tuberculosis* (Mtb) to short- or medium-chain fatty acids or cholesterol.

**Figure supplement 2.** [1–$^{14}$C]-oleic acid uptake kinetic data.

**Figure supplement 2—source data 1.** Spreadsheet of [1-$^{14}$C]-oleic acid uptake rates in indicated strains.

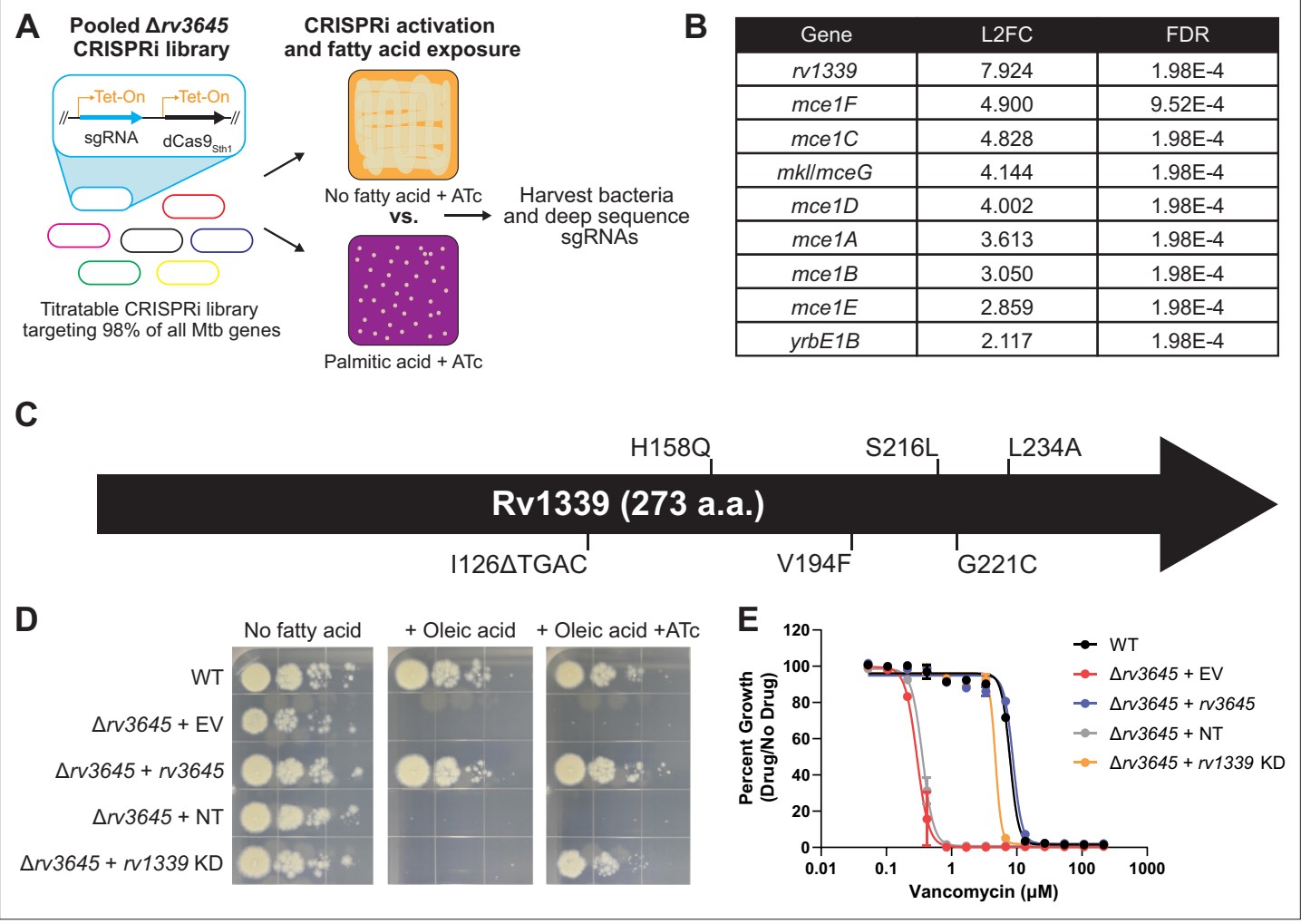

**Figure 4.** Loss-of-function of the atypical cAMP phosphodiesterase Rv1339 rescues fatty acid and drug sensitivity phenotypes of Δ*rv3645* strains. (**A**) Schematic of the Δ*rv3645* CRISPRi suppressor screen. First, an inducible genome-wide CRISPRi library was cloned into Δ*rv3645* Mtb. The CRISPRi library was then expanded before plating on 7H10-ADC agar supplemented with anhydrotetracycline (ATc) in the presence or absence of an inhibitory concentration of palmitic acid (200 μM). Genomic DNA from surviving bacteria was prepared for sgRNA deep sequencing to identify genes whose inhibition permitted the growth of an Δ*rv3645* strain in the presence of palmitic acid. (**B**) List of all enriched hit genes (log2 fold change (L2FC)>2 and false discovery rate (FDR)<0.01) from the suppressor screen described in panel (**A**). (**C**) Spontaneous suppressors of Δ*rv3645* oleic acid sensitivity were isolated and genomes were sequenced. Identified mutations in *rv1339* are shown. (**D**) Growth of Δ*rv3645* CRISPRi suppressor strains. EV = empty vector; NT = non-targeting sgRNA; KD = knockdown. (**E**) Vancomycin dose-response curves of the indicated Δ*rv3645* strains. Data represent mean ± SEM for technical triplicates and are representative of at least two independent experiments.

The online version of this article includes the following source data for figure 4:

**Source data 1.** MAGeCK screen hits and results.

shown to be an importer of fatty acids, including palmitic acid, in Mtb (*Nazarova et al., 2017*). Thus, Δ*rv3645* Mce1 knockdown strains likely fail to import palmitic acid, thereby allowing growth in the presence of this fatty acid. The fact that knockdown of the Mce1 transporter allows Δ*rv3645* to grow in the presence of palmitic acid suggests that the fatty acid toxicity phenotype of Δ*rv3645* is dependent on palmitic acid uptake and metabolism, consistent with recent results (*Fieweger et al., 2023*), rather than an uptake-independent toxicity mechanism such as fatty acid-dependent disruption of the Mtb envelope (*Kengmo Tchoupa et al., 2022*).

The top hit in the CRISPRi suppressor screen was the non-essential gene *rv1339* (*Figure 4B*). Consistent with the loss of Rv1339 activity suppressing the fatty acid sensitive phenotype of an Δ*rv3645* strain, isolation of spontaneous suppressors of Δ*rv3645* grown in the presence of oleic acid identified five unique mutations in *rv1339*, including one frameshift mutation (*Figure 4C*). We confirmed that

CRISPRi knockdown of *rv1339* rescued Δ*rv3645* fatty acid sensitivity with individual strains (*Figure 4D*). To test whether loss of Rv1339 also suppresses the fatty acid-dependent drug sensitivity phenotype, we performed MIC assays in an Δ*rv3645 rv1339* knockdown strain. *rv1339* knockdown rescued the vancomycin sensitivity phenotype of the Δ*rv3645* strain (*Figure 4E*).

Intriguingly, Rv1339 was recently reported to be an atypical cAMP phosphodiesterase (*Thomson et al., 2022*). While Rv3645 is the sole essential adenylate cyclase in H37Rv, it is possible that one or more of the other 14 additional adenylate cyclase homologs encoded in the genome (*Figure 1—figure supplement 1A, B*) could also be synthesizing cAMP under these growth conditions. In this case, the knockdown of the cAMP-degrading enzyme Rv1339 could restore cAMP levels in an Δ*rv3645* strain. These results strongly implicate cAMP levels in coordinately regulating fatty acid metabolism and intrinsic drug resistance in Mtb.

## The second messenger cAMP is a critical mediator of fatty acid metabolism and multidrug intrinsic resistance in Mtb

To test the hypothesis that cAMP regulates fatty acid metabolism and intrinsic drug resistance in Mtb, we first sought to test whether *rv3645* mutants incapable of synthesizing cAMP could complement these phenotypes. We cloned an *rv3645* allele with a point mutation in a metal-coordinating residue known to be essential for adenylate cyclase catalytic activity (*Linder et al., 2002*). The Rv3645 adenylate cyclase catalytic mutant was unable to complement oleic acid and vancomycin sensitivity phenotypes (*Figure 5A and B*). We confirmed by western blot that the catalytic mutant expressed at wild-type levels, and thus we attribute the lack of complementation specifically to the loss of cAMP synthesis (*Figure 5—figure supplement 1*).

To validate that loss of Rv3645 reduces intracellular cAMP levels in Mtb grown in 7H9-OADC, we quantified cAMP levels by mass spectrometry. A 31-fold reduction in cAMP was observed in the *rv3645* deletion strain (*Figure 5C*) that could be rescued by the knockdown of *rv1339* (*Figure 5D*). Conversely, overexpression of *rv1339* in an otherwise wild-type background led to an 11-fold reduction in cAMP levels (*Figure 5E*). Overexpression of a catalytically dead *rv1339* allele did not reduce cAMP levels (*Figure 5E*; *Thomson et al., 2022*). Reduction in cAMP levels in Δ*rv3645* and *rv1339* overexpressing strains did not alter levels of ATP nor pyrophosphate, involved in cAMP biosynthesis, indicating that the observed phenotypes are not caused by alterations in these metabolites (*Figure 5—figure supplement 2*).

Consistent with the role of cAMP in regulating long-chain fatty acid metabolism and drug sensitivity, overexpression of *rv1339* resulted in elevated oleic acid uptake and metabolism and increased vancomycin sensitivity (*Figure 5F–H*). Elevated oleic acid uptake and metabolism were not mediated by changes in Mce1 protein expression levels (*Figure 5—figure supplement 3*). To conclusively demonstrate that modulating cAMP levels is sufficient to regulate Mtb H37Rv antibiotic sensitivity, we treated Δ*rv3645* strains with V-59, a small molecule agonist of the adenylate cyclase Rv1625c that results in constitutive cAMP production (*Figure 5—figure supplement 4*; *Wilburn et al., 2022*). Addition of V-59 rescued the vancomycin sensitivity of Δ*rv3645* (*Figure 5I*). These results demonstrate the crucial role of cAMP in regulating long-chain fatty acid metabolism and multidrug intrinsic resistance in Mtb.

Rv3645 is the dominant source of cAMP under standard laboratory growth conditions (*Figure 5C*). To test whether Rv3645 is necessary to elevate cAMP levels in response to long-chain fatty acids or antibiotics, we quantified cAMP levels in Δ*rv3645* and *rv1339* overexpressing strains exposed to long-chain fatty acids and/or antibiotics. The addition of oleic acid did not alter cAMP levels as compared to fatty-acid-free media (*Figure 5—figure supplement 5A*). While the presence of antibiotics variably increased cAMP levels (*Figure 5—figure supplement 5B*), there was no correlation between the drugs that result in elevated cAMP levels and those drugs that *rv3645*-KD is sensitized to (*Figure 1B–D*). Taken together, these results suggest that Rv3645 is not a detector of long-chain fatty acids or antibiotics, but rather that cAMP produced by Rv3645 puts the bacilli in a physiologic state capable of surviving these stresses.

Lastly, to determine whether Rv3645 plays a similar role in regulating fatty acid metabolism and multidrug intrinsic resistance beyond the reference Mtb strain H37Rv, we investigated *rv3645* in the lineage 2 strain HN878. Unlike H37Rv, *rv3645* was not essential for in vitro growth in standard lab culture conditions, consistent with prior screening results (*Bosch et al., 2021*; *Carey et al., 2018*).

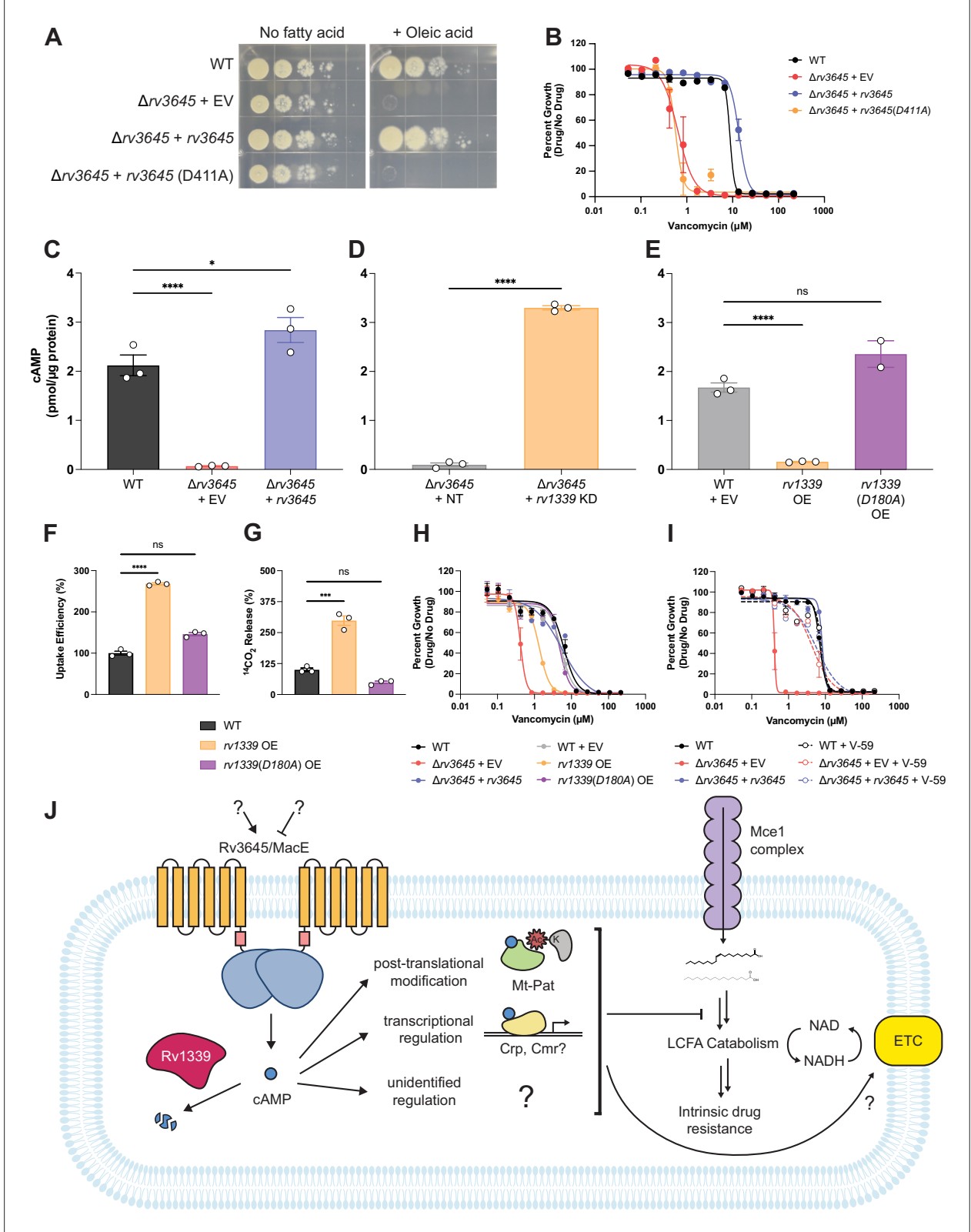

**Figure 5.** The second messenger cyclic AMP (cAMP) is a critical mediator of fatty acid metabolism and multidrug intrinsic resistance in *Mycobacterium tuberculosis* (Mtb). (**A**) Growth on 7H10-ADC +/- oleic acid of indicated Δ*rv3645* strains expressing an *rv3645* adenylate cyclase catalytic mutant (D411A). (**B**) Vancomycin dose-response curves of indicated CRISPRi strains with adenylate cyclase *rv3645(D411A)* catalytic mutant defective in cAMP production. EV = empty vector. Data represent mean ± SEM for technical triplicates. (**C–E**) cAMP measurement of the indicated strains. D180A is a catalytically

*Figure 5 continued*

dead *rv1339* allele. NT = non-targeting sgRNA; KD = knockdown; OE = over-expressed. Data represent mean ± SEM for technical triplicates. ns = not significant; *, p<0.05; ****, p<0.0001. Statistical significance was assessed by one-way ANOVA (GraphPad Prism). (**F**) Uptake of [1-$^{14}$C]-oleic acid in indicated strains. Uptake rates were calculated from the incorporated radioactive counts (***Figure 3—figure supplement 2***). Statistical significance was determined by one-way ANOVA. (**G**) Catabolic release of $^{14}CO_2$ from [1-$^{14}$C]-oleic acid in the indicated strains. Data are normalized to cell number as estimated by $OD_{600}$, quantified relative to WT, represent means ± SEM from technical triplicates, and are representative of two independent experiments. OE = over-expression, ns = not significant; ***, p<0.001; ****, p<0.0001. Statistical significance was determined by one-way ANOVA. (**H**) Vancomycin dose-response curves of *rv3645* deletion mutants overexpressing *rv1339*. (**I**) Vancomycin dose-response curves of the indicated Δ*rv3645* strains grown in the presence (dotted lines) or absence (solid lines) of the adenylate cyclase Rv1625c agonist V-59. Data represent mean ± SEM for technical triplicates and are representative of at least two independent experiments. (**J**) Model for the involvement of cAMP in lipid metabolism and intrinsic drug resistance in Mtb. Under standard lab culture conditions (7H9/7H10 media), Rv3645/MacE is the dominant source of cAMP. cAMP regulates physiological processes through binding to transcription factors, post-translational modification enzymes, and other poorly understood effector proteins. Regulation through likely multiple of these effector proteins reduces long-chain fatty acid uptake and catabolism and promotes intrinsic multidrug resistance.

The online version of this article includes the following source data and figure supplement(s) for figure 5:

**Figure supplement 1.** *rv3645* catalytic mutant is expressed similar to the wild-type allele.

**Figure supplement 1—source data 1.** Original Western blot image of *rv3645* catalytic mutant.

**Figure supplement 2.** ATP and pyrophosphate levels does not change in Δ*rv3645* or *rv1339* overexpression strains.

**Figure supplement 3.** Expression of Mce1 complex proteins does not change in Δ*rv3645* or *rv1339* overexpression strains.

**Figure supplement 3—source data 1.** Original Western blot images of Mce1 complex proteins and GroEL.

**Figure supplement 4.** V-59 induces cyclic AMP (cAMP) production in Δ*rv3645*.

**Figure supplement 5.** An inducible cyclic AMP (cAMP) response is not critical to modulate oleic acid sensitivity nor intrinsic drug resistance.

**Figure supplement 6.** Depletion of cyclic AMP (cAMP) sensitizes Mtb HN878 to vancomycin and palmitic acid.

**Figure supplement 7.** Δ*macE* is hypersensitive to the cytochrome $bc_1$ complex inhibitor Q203.

Despite this difference in essentiality, the reduction of cAMP levels sensitized HN878 to both vancomycin and palmitic acid (***Figure 5—figure supplement 6***). Taken together, these data are consistent with a critical role for cAMP in regulating fatty acid metabolism and intrinsic drug resistance in Mtb.

## Discussion

Our work defines *rv3645* and cAMP as central mediators of intrinsic multidrug resistance and fatty acid metabolism in Mtb H37Rv. *rv3645* knockdown resulted in increased sensitivity to antibiotics with diverse targets by a mechanism largely independent of increases in cell envelope permeability. Surprisingly, we found that *rv3645* was only essential in H37Rv in the presence of long-chain fatty acids. Suppression of long-chain fatty acid sensitivity was conferred by loss of the Mce1 transporter, presumably reflecting reduced fatty acid uptake, or inactivation of the atypical cAMP phosphodiesterase Rv1339. Consistent with these genetic results, using mass spectrometry we found that Rv3645 is the dominant source of cAMP under standard laboratory conditions in H37Rv, despite this strain encoding 14 additional adenylate cyclases in its genome. We propose naming Rv3645 "MacE" for the <u>m</u>ajor <u>a</u>denylate <u>c</u>yclase <u>e</u>nzyme. Using gain- and loss-of-function alleles and small molecule-regulated cAMP production, we show that reduced cAMP levels are associated with elevated oleic acid uptake and metabolism, long-chain fatty acid toxicity, and increased drug sensitivity.

MacE is predicted to be composed of six transmembrane helices and cytosolic HAMP and cyclase homology domains. The presence of a HAMP domain, which typically transmits conformational changes from a periplasmic or transmembrane ligand binding domain to a cytoplasmic signaling domain (***Hulko et al., 2006***), suggests that MacE senses a ligand and transduces this signal into cytoplasmic cAMP production. The nature of this signal remains to be determined, but it is unlikely to be long-chain fatty acids or antibiotics. Rather, our data suggest that MacE is active in the presence or absence of long-chain fatty acids and antibiotics, and that cAMP produced by MacE puts the bacilli in a physiologic state capable of surviving these stresses.

Why does a lack of *macE* and lowered cAMP levels make Mtb more sensitive to long-chain fatty acids and drugs? Prior work established a link between cAMP and fatty acid catabolism through the cAMP-activated lysine acetyltransferase Mt-Pat (***Nambi et al., 2013***). Activated Mt-Pat acetylates/

propionylates multiple enzymes involved in fatty acid catabolism, including ten FadD paralogs and acetyl-CoA synthetase, thereby inhibiting AMP-ligase activity and oxidation of fatty acids to acetyl-CoA. Inactivation of Mt-Pat, like *macE*, makes Mtb more sensitive to fatty acids (*Nambi et al., 2013*; *Rittershaus et al., 2018*), although the fatty acid sensitivity phenotype is more severe for loss of *macE* as evidenced by the non-essentiality of Mt-Pat under standard laboratory culture conditions (*DeJesus et al., 2017*). Further linking cAMP and fatty acid metabolism, VanderVen and colleagues showed that transposon mutants in *rv1339* showed reduced uptake of BODIPY-palmitate in infected macrophages (*Nazarova et al., 2019*). Collectively, growing evidence suggests that reduced cAMP levels lead to overactive fatty acid catabolism.

In a series of related findings, Sassetti and colleagues screened Mtb transposon mutant libraries and identified Mt-Pat mutants as strongly attenuated under hypoxia (*Rittershaus et al., 2018*). In the absence of Mt-Pat, it was hypothesized that Mtb fails to downregulate fatty acid catabolism under hypoxic conditions, leading to a continual flux of acetyl-CoA through oxidative TCA metabolism. Consistent with this interpretation, the same screen identified that transposon mutants in the Mce1 transporter promoted fitness in hypoxia. Under hypoxia, the TCA cycle is thought to preferentially work in the reductive direction to regenerate NAD and prevent the accumulation of NADH in the absence of a terminal electron acceptor (*Eoh and Rhee, 2013*; *Watanabe et al., 2011*). As both the oxidative branch of the TCA cycle and fatty acid β-oxidation produce NADH, Δ*mt-pat* under hypoxic conditions becomes redox imbalanced and depletes NAD. Also potentially consistent with this interpretation, re-analysis of the hypoxia TnSeq data identifies *rv1339* as the top resistance-promoting hit (*Rittershaus et al., 2018*). Elevated cAMP levels in *rv1339* transposon mutants could augment Mt-Pat activity and reduce fatty acid uptake and catabolism and drive expression of malate dehydrogenase (*Gazdik and McDonough, 2005*), an enzyme critical for the reductive TCA cycle under hypoxia (*Rittershaus et al., 2018*).

Finally, recent observations suggest a link between cAMP and the electron transport chain. Isolation of spontaneous resistant mutants to a series of novel inhibitors of the QcrB subunit of the cytochrome $bc_1$-$aa_3$ oxidase repeatedly identifies mutations in *rv1339* (*Chandrasekera et al., 2017*; *O'Malley et al., 2018*; *Shelton et al., 2021*). How loss of *rv1339* and elevated cAMP promotes resistance to QcrB inhibitors remains to be defined, but in principle could be achieved by elevated cAMP levels increasing electron transport chain activity, for example by increasing expression of the alternative terminal oxidase cytochrome *bd* (*Ko and Oh, 2020*). Indeed, Δ*macE* shows hypersensitivity to the QcrB inhibitor Q203 (*Figure 5—figure supplement 7*), potentially suggesting a cytochrome *bd* defect in this strain.

Collectively, ours and published results lead to the following working model (*Figure 5J*). Loss of *macE* reduces cAMP levels which reduces the activity of multiple cAMP effectors, including Mt-Pat. That no single annotated or predicted cAMP effector individually recapitulates the phenotypes observed with *macE* indicates that the physiological effects of reduced cAMP levels in Δ*macE* are the consequence of the altered activity of multiple effectors (*Figure 1—figure supplement 1A*). Potential downstream cAMP effector proteins include transcription factors like CRP, transporters, phospholipases, and others (*Johnson and McDonough, 2018*), highlighting the potential pleiotropic consequences of modulating cAMP levels. Reduced activity of cAMP-responsive proteins, including Mt-Pat, may increase fatty acid uptake and catabolism (*Nazarova et al., 2019*) and oxidative TCA metabolism while simultaneously reducing electron transport chain activity (*Ko and Oh, 2020*). This derangement may ultimately lead to redox imbalance, perturbed membrane potential, reduced growth, and increased sensitivity to antibiotics through yet-to-be-defined mechanisms.

Much work remains to be done to test the predictions generated by this model. The lack of changes in Mce1 protein abundance (*Figure 5—figure supplement 3*) despite increased fatty acid uptake in low cAMP strains may indicate complex regulation of fatty acid uptake, including post-translational modifications or activity-modulating accessory proteins of the Mce1 transporter. Moreover, it is known that host-associated signals like low pH can activate Mtb adenylate cyclases and that macrophage infection results in a burst of bacterial cAMP production (*Bai et al., 2009*; *Tews et al., 2005*). Thus, it is likely that some of the up to 14–15 additional adenylate cyclases encoded in the Mtb genome are active in the diverse niches encountered by Mtb during infection. Whether cAMP produced under these diverse host-associated conditions also coordinates fatty acid metabolism and intrinsic multidrug resistance remains to be investigated. It is interesting to note that both *macE* and *rv1339* were

recently found to be under positive selection (*Liu et al., 2022*), raising the possibility that alterations in cAMP levels are being positively selected in Mtb clinical isolates.

Notably, the canonical cAMP phosphodiesterase *rv0805* was not identified as a suppressor of palmitic acid toxicity in the Δ*macE* suppressor screen nor in the hypoxia TnSeq screen (*Rittershaus et al., 2018*). Rv0805 is 150 times more active against 2',3'-cAMP, a cAMP species associated with RNA degradation, than 3',5'-cAMP, the product of adenylate cyclases (*Keppetipola and Shuman, 2008*). Consistent with these data, overexpression of Rv0805 showed only a modest (~40%) reduction in 3',5'-cAMP in Mtb (*Agarwal et al., 2009*). Lastly, *rv0805* has a limited phylogenetic distribution and is not found in *M. smegmatis*, which also encodes numerous adenylate cyclases. Thus, it appears that *rv1339* and not *rv0805* is the major 3',5'-cAMP phosphodiesterase in Mtb (*Thomson et al., 2022*).

We show here that the sole in vitro essential adenylate cyclase in H37Rv, MacE, links fatty acid metabolism and intrinsic multidrug resistance through the production of cAMP. The adenylate cyclase Rv1625c has recently been shown to be important for cholesterol catabolism, although interestingly this phenotype was independent of the Rv1625c cyclase homology domain (*Wilburn et al., 2022*). In a screen for compounds that disrupt cholesterol catabolism, VanderVen and colleagues discovered V-59, an Rv1625c agonist analogous to the eukaryotic adenylate cyclase activator forskolin (*VanderVen et al., 2015*; *Wilburn et al., 2022*). V-59 results in constitutive activation of Rv1625c and cAMP production and, for reasons yet unknown, blocks cholesterol catabolism. Thus, it would appear proper metabolic function of Mtb grown on fatty acids and cholesterol requires 'just the right amount' of cAMP: too little and Mtb cannot grow in the presence of fatty acids, too much and Mtb cannot catabolize cholesterol. Moreover, our results and those investigating combination drug treatment with V-59 (*Wilburn et al., 2022*) suggest that deficient or elevated cAMP production may potentiate combination drug therapy.

Interestingly, whereas cAMP promotes intrinsic multidrug resistance in Mtb, this relationship may not be conserved or in some cases may even be reversed in other bacteria. Large-scale chemical genomic screening of cAMP-deficient Δ*cya E. coli* did not identify any significant differences in antibiotic susceptibility (*Nichols et al., 2011*), although earlier studies found that Δ*cya* in *E. coli* and *S. typhimurium* promotes fosfomycin resistance by reducing expression of the GlpT and UhpT uptake systems (*Shiver et al., 2016*; *Silver, 2017*). In uropathogenic *E. coli*, cAMP was also found to be a negative regulator of persistence (*Molina-Quiroz et al., 2018*). Δ*cya* upregulated the oxidative stress response and SOS-dependent DNA damage repair and promoted survival to beta-lactam antibiotics. Thus, it will be interesting to examine the relationship between cAMP and drug efficacy across diverse bacterial species.

cAMP is a ubiquitous but poorly understood second messenger in Mtb. Mtb devotes a considerable amount of coding capacity to produce, sense, and degrade cAMP. Here, we reveal the adenylate cyclase MacE as the dominant source of cAMP under standard laboratory growth conditions. cAMP levels are coordinately regulated by MacE and the atypical phosphodiesterase Rv1339. cAMP produced by MacE is critical for Mtb growth on long-chain fatty acids, a host-relevant carbon source, and for intrinsic multidrug resistance, highlighting the potential utility of small molecule modulators of this second messenger to control Mtb infection (*Wilburn et al., 2022*).

## Materials and methods

### Bacterial strains

Mtb strains are derivatives of H37Rv, unless indicated as HN878. *E. coli* strains are derivatives of DH5alpha (NEB). *M. smegmatis* strains are derivatives of mc$^2$155 *groEL1ΔC* (*Noens et al., 2011*). Resources used to construct bacterial strains are listed in *Supplementary file 1*.

### Mycobacterial cultures

Mtb was grown at 37 °C in Difco Middlebrook 7H9 broth or on 7H10 agar supplemented with 0.2% glycerol (7H9) or 0.5% glycerol (7H10), 0.05% Tween-80, 1x oleic acid-albumin-dextrose-catalase (OADC) and the appropriate antibiotics, unless otherwise specified. Media for the Δ*macE* strain and strains to be tested for fatty acid sensitivity or fatty acid-dependent phenotypes were similarly prepared except 0.05% tyloxapol was used instead of Tween-80, and fatty acid-free albumin-dextrose-catalase (ADC) was used instead of OADC. Where required, antibiotics or small molecules were used

at the following concentrations: kanamycin at 20 µg/mL; anhydrotetracycline (ATc) at 100 ng/mL, hygromycin at 50 µg/mL, zeocin at 20 µg/mL, and V-59 at 10 µM. Mtb cultures were grown standing in tissue culture flasks (unless otherwise indicated) at 37 °C, 5% $CO_2$. Fatty acid sensitivity testing on 7H10 agar was conducted with 500 µM oleic acid or 200 µM palmitic acid.

*M. smegmatis* was grown at 37 °C in similarly supplemented 7H9 broth or 7H10 agar except ADC was used instead of OADC.

## Total RNA extraction and RNA-seq

Triplicate cultures were grown to mid-log phase in 7H9-ADC and diluted back to $OD_{600}$ 0.2 in 7H9-OADC. Cultures were incubated for 48 hr. Total RNA extraction was performed as previously described (*Bosch et al., 2021*). Briefly, 2 $OD_{600}$ units of bacteria were added to an equivalent volume of GTC buffer (5 M guanidinium thiocyanate, 0.5% sodium N-lauryl sarcosine, 25 mM trisodium citrate dihydrate, and 0.1 M 2-mercaptoethanol), pelleted by centrifugation, resuspended in 1 mL TRIzol (Thermo Fisher 15596026) and lysed by zirconium bead beating (MP Biomedicals 116911050). Chloroform (0.2 ml) was added to each sample and samples were frozen at −80 °C. After thawing, samples were centrifuged to separate phases and the aqueous phase was purified by Direct-zol RNA miniprep (Zymo Research R2052). Residual genomic DNA was removed by TURBO DNase treatment (Invitrogen Ambion AM2238). RNA was purified and concentrated (Zymo Research R1017) and sent to SeqCenter (Pittsburgh, PA, USA) for library preparation. Samples were DNAse treated with Invitrogen DNAse (RNAse free). Library preparation was performed using Illumina's Stranded Total RNA Prep Ligation with Ribo-Zero Plus kit and 10 bp IDT for Illumina indices. Sequencing was done on a NextSeq2000 giving 2 × 51 bp reads. Demultiplexing, quality control, and adapter trimming were performed with bcl-convert (v3.9.3).

## Processing and analysis of RNA-seq data

Raw FASTQ files were aligned to the H37Rv genome (NC_018143.2) using Rsubread (version 2.0.1) (*Liao et al., 2019*) with default settings. Transcript abundances were calculated by processing the resulting BAM files with the summarizeOverlaps function of the R package GenomicAlignments (version 1.22.1) (*Lawrence et al., 2013*). Overlaps were calculated in the 'Union' mode, ensuring reads were counted only if they overlap a portion of a single gene/feature. 16 S, 23 S, and 5 S rRNA features (RVBD6018, 6019, and 6020, respectively) were manually removed from the count data to prevent confounding downstream differential gene expression analysis. Reads per kilobase of transcript, per million reads mapped (RPKM) determined for each gene by normalizing transcript abundance to gene length.

## Domain prediction

MacE domain prediction was conducted using the Conserved Domain Database (https://www.ncbi.nlm.nih.gov/Structure/cdd/wrpsb.cgi). Transmembrane helices were predicted using TMHMM (https://services.healthtech.dtu.dk/service.php?TMHMM-2.0).

## Antibacterial activity and fatty acid sensitivity measurements

All antibiotics were dissolved in DMSO (VWR V0231) and dispensed using an HP D300e Digital Dispenser in a 384-well plate format. DMSO did not exceed 1% of the final culture volume and was maintained at the same concentration across all samples. CRISPRi strains were growth-synchronized and pre-depleted in the presence of ATc (100 ng/mL) for 4 days prior to assay for MIC analysis. Cultures were then back diluted to a starting $OD_{600}$ of 0.05 in 7H9-OADC, and 50 µL of cell suspension was plated in technical triplicate in wells containing the test compound and fresh ATc (100 ng/mL). Δ*macE* strains were cultured in 7H9-ADC prior to back diluting in 7H9-OADC to seeding a 384-well plate. Similarly, for strains used in antibacterial activity testing in the presence or absence of fatty acids, strains were grown in 7H9-ADC before back-diluting the culture to a starting $OD_{600}$ of 0.05 in 7H9-ADC or 7H9-OADC to seed a 384-well plate. Plates were incubated at 37 °C with 5% $CO_2$. $OD_{600}$ was evaluated using a Tecan Spark plate reader at 14–18 days post-plating and percent growth was calculated relative to the vehicle control for each strain. $IC_{50}$ measurements were calculated using a non-linear fit in GraphPad Prism.

For fatty acid sensitivity measurements, fatty acids were dissolved in 1:1 tyloxapol:ethanol and then diluted at 2x the maximum testing concentration in 7H9-ADC. 2-fold serial dilutions were prepared and 25 µL of each concentration was transferred to a 384-well plate. Cultures were grown to $OD_{600}$ 0.4–0.6. Cultures were then back diluted to a starting $OD_{600}$ of 0.1 and 25 µL of cell suspension was plated in technical triplicate in wells containing 25 µL of the fatty acid dilution series (and fresh ATc (100 ng/mL), where applicable).

To quantify growth phenotypes on 7H10 agar, 10-fold serial dilutions of OD-synchronized Mtb cultures were spotted on 7H10-ADC agar containing fatty acids at the indicated concentrations. Where applicable, ATc was added at 100 ng/mL. Plates were incubated at 37 °C and imaged after two weeks.

## Cell wall permeability assay

Cell envelope permeability was determined using the ethidium bromide (EtBr) uptake assay as previously described (*Xu et al., 2017*). Briefly, mid-log-phase Mtb cultures were washed once in PBS +0.05% Tween-80 (PBST) and adjusted to $OD_{600}$ 0.8 in PBST supplemented with 0.4% glucose. 100 µL of bacterial suspension was added to a black 96-well clear-bottomed plate (Costar). After this, 100 µL of 8 µg/mL EtBr in PBST supplemented with 0.4% glucose was added to each well. EtBr fluorescence was measured (excitation: 530 nm/emission: 590 nm) at 1 min intervals over a course of 90 min. EtBr fluorescence at 30 min is plotted, and normalized to optical density. Experiments were performed in technical triplicate. Similar uptake assays were performed with Calcein-AM (Invitrogen, #C3099) and BCECF-AM (Invitrogen #B1150) with the following changes (1) Calcein-AM and BCECF-AM were assayed at a final concentration of 1 µg/mL. (2) Calcein fluorescence was measured with the following parameters (excitation: 495 nm/emission: 520 nm) (3) BCECF fluorescence was measured with the following parameters (excitation: 490 nm/emission: 530 nm). Plotting and normalization were performed similarly to EtBr.

A similar assay was performed to determine envelope permeability to a fluorescent vancomycin analog, except that: (1) the bacterial suspension was adjusted at $OD_{600}$ = 0.4 in PBS supplemented with 0.4% glucose; (2) cells were incubated with 2 µg/mL BODIPY FL Vancomycin (Thermo Scientific, V34850) for 30 min; (3) 900 µL sample aliquots were taken at different time points, washed twice with PBS, resuspended in 600 µL PBS, and three aliquots of 200 uL each were transferred to a black 96-well clear-bottomed plate (Costar); and (4) fluorescence was measured (excitation: 485 nm/emission: 538 nm) and normalized to the $OD_{600}$ of the final bacterial suspension.

## Vancomycin time-kill kinetic assay

CRISPRi strains were growth-synchronized and pre-depleted in the presence of ATc (100 ng/mL) for 4 days prior to assay for kill kinetic analysis. Cultures were then back diluted to a starting $OD_{600}$ of 0.1 in 7H9-OADC with 1.16 µM vancomycin, 100 ng/mL ATc, and incubated at 37 °C for 96 hr. Bacterial counts for each strain were determined at 0, 0.5, 6, 24, and 96 hr post-antibiotic exposure by plating serial 10-fold dilutions of each culture on 7H10-ADC. Colonies were counted after 17 days of incubation at 37 °C.

## Measuring drug uptake by mass-spectrometry

CRISPRi strains were growth-synchronized and pre-depleted in the presence of ATc (100 ng/mL) for 4 days. Growth in 7H9-OADC ATc media proceeded to an OD of 0.8–1.0. One OD unit was used to seed filters for growth on 7H10-OADC ATc agar plates. After 5 days of growth, bacteria-laden filters were transferred and floated on 7H9-OADC ATc media in a 'swimming pool' set up overnight. Filters were then transferred and floated on drug-containing 7H9-OADC ATc swimming pools and incubated for 24 hr. Media was collected from each pool and filter sterilized using 0.22 micron nylon centrifugal filters (Corning). 100 µL of each media sample was added to 400 µL of 1:1 acetonitrile/methanol and centrifuged at 13,000 *g* for 10 min at 4 °C to pellet precipitated protein. Supernatants were transferred into mass spectrometry vials and analyzed using a semi-quantitative LC/MS-based method as described previously (*Planck and Rhee, 2021*). Briefly, samples were separated on a Cogent Diamond Hydride Type C column (Microsolv Technologies). The mobile phase consisted of solvent A (ddH$_2$O with 0.2% formic acid) and solvent B (acetonitrile with 0.2% formic acid), and the gradient used was as follows: 0–2 min, 85% B; 3–5 min, 80% B; 6–7 min, 75% B; 8–9 min, 70% B; 10–11.1 min, 50% B;

11.1–14 min 20% B; 14.1–24 min 5% B, followed by a 10 min re-equilibration period at 85% B at a flow rate of 0.4 mL/min. This was achieved using an Agilent 1200 Series liquid chromatography (LC) system coupled to an Agilent 6546 quadrupole time of flight (Q-TOF) mass spectrometer in positive acquisition mode, and 2 µL of the sample were injected for each run. Dynamic mass axis calibration was achieved by continuous infusion of a reference mass solution using an isocratic pump with a 100:1 splitter. Resulting data were analyzed using Agilent MassHunter Qualitative Analysis Navigator software. Relative antibiotic concentrations were determined by quantitation of peak heights using $m/z$ of 338.1511 for the linezolid $(M+H)^+$ ion (RT = 1.3 min) and 724.7224 for the vancomycin $(M+2 H)^{2+}$ ion (RT = 8.8 min) with a mass tolerance of +/-30 ppm. To determine uptake, drug levels in bacteria-laden filter swimming pools were compared to levels from control swimming pools incubated with cell-free filters under the same conditions.

## Generation of individual CRISPRi and CRISPRi-resistant complementation strains

Individual CRISPRi plasmids were cloned as previously described in *Bosch et al., 2021* using Addgene plasmid #166886. Briefly, the CRISPRi plasmid backbone was digested with BsmBI-v2 (NEB #R0739L) and gel purified. sgRNAs were designed to target the non-template strand of the target gene ORF. For each individual sgRNA, two complementary oligonucleotides with appropriate sticky end overhangs were annealed and ligated (T4 ligase NEB # M0202M) into the BsmBI-digested plasmid backbone. Successful cloning was confirmed by Sanger sequencing.

Individual CRISPRi plasmids were then electroporated into Mtb. Electrocompetent cells were obtained as described in *Murphy et al., 2015*. Briefly, a WT Mtb culture was expanded to an $OD_{600}$ = 0.8–1.0 and pelleted (4000 × g for 10 min). The cell pellet was washed three times in sterile 10% glycerol. The washed bacilli were then resuspended in 10% glycerol in a final volume of 5% of the original culture volume. For each transformation, 100 ng plasmid DNA and 100 µL of electrocompetent mycobacteria were mixed and transferred to a 2 mm electroporation cuvette (Bio-Rad #1652082). Where necessary, 100 ng of plasmid pIRL19 (Addgene plasmid #163634) was also added. Electroporation was performed using the Gene Pulser X cell electroporation system (Bio-Rad #1652660) set at 2500 V, 700 Ω, and 25 µF. Bacteria were recovered in 7H9 for 24 hr. After the recovery incubation, cells were plated on 7H10 agar supplemented with the appropriate antibiotic to select for transformants.

To complement CRISPRi-mediated gene knockdown, synonymous mutations were introduced into the complementing allele at both the protospacer adjacent motif (PAM) and seed sequence (the 8–10 most PAM-proximal bases at the 3' end of the sgRNA targeting sequence) to prevent sgRNA targeting, as described here (*Wong and Rock, 2021*). Silent mutations were introduced into Gibson assembly oligos to generate these 'CRISPRi resistant' (CR) alleles. Complementation alleles were expressed from hsp60 promoters in a Tweety integrating plasmid backbone, as indicated in each figure legend and/or the relevant plasmid maps (*Supplementary file 1*). These alleles were then transformed into the corresponding CRISPRi knockdown strain.

The full list of sgRNA targeting sequences and complementation plasmids can be found in *Supplementary file 1*.

## Construction of the Δ*macE* and complemented Mtb strains

Mtb H37Rv gene *rv3645/macE* was deleted through homologous recombination using a strain expressing the recombinase RecET (*Murphy et al., 2015*). To induce RecET expression, isovaleronitrile was added at a final concentration of 1 µM to a mid-exponential Mtb culture (optical density 580 nm of approximately 1) for 8 hr after which glycine was added at a final concentration of 2 M, and the culture was incubated overnight. A construct composed of a hygromycin-resistant gene (hygR) flanked by 500 bp upstream and downstream of *macE* was synthesized (GenScript) and electroporated into Mtb expressing the recombinase RecET. A *macE* deletion mutant strain (Δ*macE*) was selected in solid fatty acid-free modified Sauton's with hygromycin. The plasmid expressing recET (pNitET-SacB-kan) was counter selected by growing the deleted mutant in solid fatty acid-free modified Sauton's supplemented with sucrose 10%. For complementation of Δ*macE*, we have cloned *macE* under the control of the promoter Phsp60 into a plasmid with a kanamycin-resistant cassette that integrates at the att-L5 site (pMCK-Phsp60-rv3645) and electroporated it into the *macE* deletion mutant. Primers and primers are listed in *Supplementary file 1*.

## Isolation of spontaneous fatty acid toxicity suppressors

The mutant strain Δ*macE* was grown in fatty acid-free modified Sauton's minimal medium until stationary phase (*Beites et al., 2019*). Solid fatty acid-free modified Sauton's minimal medium supplemented with oleic acid at a final concentration of 500 µM was used to select spontaneous mutants in the Δ*macE* genetic background that regained the ability to grow in the presence of oleic acid. We inoculated this medium with $10^7$ and $10^8$ bacteria and incubated the plates for 4 weeks. Medium not supplemented with oleic acid was used as a viability control. Colonies that grew in the medium supplemented with oleic acid were picked and grown in liquid fatty acid-free modified Sauton's minimal medium. To validate the isolated rescue mutants, we cultured WT, Δ*macE,* complemented, and rescue mutants in liquid fatty acid-free modified Sauton's minimal medium supplemented with a concentration of oleic acid restrictive to Δ*macE* growth.

## Whole genome sequencing

The genetic identity of Δ*macE* and derived spontaneous suppressor mutants was confirmed by whole genome sequencing (WGS). Genomic DNA (150–200 ng) was sheared and HiSeq sequencing libraries were prepared using the KAPA Hyper Prep Kit (Roche). Libraries were amplified by PCR (10 cycles). $5–10 \times 10^6$ 50 bp paired-end reads were obtained for each sample on an Illumina HiSeq 2500 using the TruSeq SBS Kit v3 (Illumina). Post-run demultiplexing and adapter removal were performed and FASTQ files were inspected using fastqc (*Andrews, 2010*). Trimmed FASTQ files and the reference genome (M. tuberculosis H37RvCO; NZ_CM001515.1) were aligned using bwa mem (*Li and Durbin, 2010*). Bam files were sorted and merged using SAMtools (*Li et al., 2009*). Read groups were added and bam files were de-duplicated using Picard tools and GATK best-practices were followed for SNP and indel detection (*DePristo et al., 2011*). Gene knockouts and cassette insertions were verified for all strains by direct comparison of reads spanning insertion points to plasmid maps and the genome sequence. Reads coverage data was obtained from the software Integrative Genomics Viewer version 2.5.2 (IGV) (*Robinson et al., 2011*).

## Construction of a genome-wide crispri library in Δ*macE*

Libraries were constructed as previously described (*Bosch et al., 2021*). Briefly, 37 transformations were performed to generate Δ*macE* RLC12 libraries. For each transformation, 1 µg of RLC12 plasmid DNA was added to 100 µL electrocompetent H37Rv Mtb cells (~$1 \times 10^{10}$ cells per transformation). The cells:DNA mix was transferred to a 2 mm electroporation cuvette (Bio-Rad #1652082) and electroporated at 2500 kV, 700 ohms, and 25 µF. Each transformation was recovered in 2 mL Sauton's media supplemented with fatty acid-free ADC, glycerol, and tyloxapol (80 mL total) for 16–24 hr. The recovered cells were harvested at 4000 rpm for 10 min, resuspended in 700 µL remaining media per transformation, and plated on Sauton's agar supplemented with kanamycin (see Bacterial cultures) in Corning Bioassay dishes (Sigma #CLS431111-16EA). Transformation efficiency was estimated from library titring and indicated >12x average sgRNA coverage of RLC12 was achieved in Δ*macE*.

After 33 days of outgrowth on plates, transformants were scraped and pooled. Scraped cells were homogenized by two dissociation cycles on a gentleMACS Octo Dissociator (Miltenyi Biotec #130095937) using the RNA_02.01 program and 10 gentleMACS M tubes (Miltenyi Biotec #130093236). The library was further declumped by passaging 10 individual *M. tuberculosis* library aliquots in 10 mL of kanamycin-supplemented Sauton's in T-25 flasks (Falcon # 08-772-1F) for 10 generations. Final Δ*macE* RLC12 library stocks were obtained after pooling the cultures and passing them through a 10 µm cell strainer (Pluriselect #SKU 43-50010-03). Genomic DNA was extracted from the final Δ*macE* RLC12 library stock and library quality were validated by deep sequencing (see Genomic DNA extraction and library preparation for Illumina sequencing).

## CRISPRi fatty acid-genetic suppressor screening

The fatty acid-genetic suppressor screen was initiated by thawing 3 × 1.5 mL aliquots (1 $OD_{600}$ unit per aliquot) of the Δ*macE* CRISPRi library and inoculating each aliquot into 8.5 mL 7H9-ADC in a vented tissue culture flask (T-25; Corning #430639). The starting $OD_{600}$ of each culture was approximately 0.1. Cultures were expanded to $OD_{600}$ = 0.47, pooled, and evenly divided to inoculate 2X90 mL cultures with 7.5 ODU each in tissue culture flasks (T-225; Falcon #353138). Cultures were expanded to OD 0.3, pooled, pelleted, and resuspended in 15 mL 7H9-ADC. 700 µL of the concentrated cells were plated

on FA-free 7H10-ADC 25 cm bioassay dishes, or 7H10-ADC with increasing concentrations of palmitic acid (200 µM) in quintuplicate. Bioassay dishes were supplemented with kanamycin at 20 µg/mL and ATc at 100 ng/mL. To titer the library, a 10-fold dilution series of the concentrated cells were plated on petri dishes with FA-free 7H10-ADC with kanamycin at 20 µg/mL. All plates were incubated for 20 days. Library coverage based on titering plates was 4620 X. Colonies from the fatty acid-containing bioassay dishes were scraped, avoiding clustered colonies, into PBS and pelleted. Due to confluent growth in the absence of selection on the fatty acid-free plates, a 3 cm × 25 cm rectangular area was scraped into PBS and cells were pelleted for genomic DNA extraction.

## Genomic DNA extraction and library preparation for illumina sequencing

Genomic DNA was isolated from bacterial pellets using the CTAB-lysozyme method as previously described (*Bosch et al., 2021*; *Larsen et al., 2007*). Genomic DNA concentration was quantified by Nanodrop. Next, the sgRNA-encoding region was amplified from 500 ng of genomic DNA using NEBNext Ultra II Q5 master Mix (NEB #M0544L). PCR cycling conditions were: 98 °C for 45 s; 17 cycles of 98 °C for 10 s, 64 °C for 30 s, 65 °C for 20 s; 65 °C for 5 min. Samples were dual-indexed. For dual-indexed samples, each PCR reaction contained a unique indexed forward and reverse primer (0.5 µM each) (*Supplementary file 1*). Forward primers contain a P5 flow cell attachment sequence, a standard Read1 Illumina sequencing primer binding site, custom stagger sequences to ensure base diversity during Illumina sequencing, and a unique barcode to allow for sample pooling during deep sequencing. Reverse primers contain a P7 flow cell attachment sequence, a standard Read2 Illumina sequencing primer binding site, and unique barcodes to allow for dual-indexed sequencing.

Following PCR amplification, each ~230 bp amplicon was purified using sparQ PureMag Beads (Quantabio # 95196–060) using double-sided size selection (first 0.75x, then an additional 0.12x for a final 0.87x). Size-selected amplicons were quantified with a Qubit 2.0 Fluorometer (Invitrogen). Amplicon size and purity were quality controlled by visualization on an Agilent 2100 Bioanalyzer (high sensitivity chip; Agilent Technologies #5067–4626). Next, individual PCR amplicons were multiplexed into 10 nM pools and sequenced on an Illumina sequencer according to the manufacturer's instructions (2.5–5% PhiX spike-in; PhiX Sequencing Control v3; Illumina # FC-110–3001). Samples were run on the Illumina NextSeq 500 platform (Single-Read 1x85 cycles and six i7 index cycles).

## Western blotting

For detection of protein expression of MacE alleles, 80 $OD_{600}$ units of growth synchronized *M. smegmatis* cultures constitutively expressing *macE* were harvested by centrifugation (4000 × g, 10 min). Cells were washed twice in 40 mL PBS-0.05% Tween80 and resuspended in 600 µL of lysis buffer (50 mM Tris, 150 mM NaCl, pH 7.4) containing a protease inhibitor cocktail (Sigma-Aldrich, #11873580001). Cells were lysed by bead beating in Lysis B Matrix tubes (MP Biomedicals; #116911050) using a Precellys Evolution homogenizer (Bertin Instruments, #P000062-PEVO0-A, 3x10,000 RPM, 30 s intervals, 4 °C). n-Dodecyl-β-D-maltopyranoside (Alfa Aesar, #J66869) was added to a final concentration of 1% and incubated at 4 °C with inversion for 2 hr. The cell lysates were cleared by centrifugation (20,000 x g, 2 min), and a 20 µL aliquot was mixed with 4x Laemmli Sample Buffer (Bio-Rad, #1610747) supplemented with DTT. Samples were separated on a 4–12% Bis-Tris polyacrylamide gel (Invitrogen, #NP0323BOX) in MOPS running buffer, transferred to a nitrocellulose membrane using the Trans-Blot Turbo Transfer System (Bio-Rad, #1704150), and incubated for 1 hr in blocking buffer (LI-COR, #927–60001). Proteins were probed with anti-RpoB (BioLegend, #663905) and anti-His (GenScript, #A00186) primary antibodies overnight at 4 °C and subsequently detected with fluorescent goat anti-mouse secondary antibodies (Bio-Rad, #12004159).

For detection of Mce1 proteins, Mtb was cultured at 37 °C in 7H9 containing fatty acid-free albumin-dextrose supplemented with 0.01% glycerol and 0.05% tyloxapol to an $OD_{600}$ of 0.6. Harvested cells were fixed with 4% PFA, washed with PBS +0.05% tyloxapol, and whole cell lysates were generated by sonicating the bacteria in 1.0% SDS. Proteins were separated using SDS-PAGE then transferred to a nitrocellulose membrane and probed with mce-specific antibodies or anti-GroEL. Anti-Mce1A, anti-Mce1D, and anti-Mce1E primary antibodies were a generous gift from Christopher Sassetti (*Feltcher et al., 2015*). Generation of anti-MceG antibodies was as described (*Nazarova et al., 2017*). GroEL antibodies were obtained from BEI resources.

## Measuring nucleotides by mass spectrometry

Strains were grown to an OD of 0.8–1.0 in 7H9-ADC media. CRISPRi strains were grown for 4 days to an OD of 0.8–1.0 in the presence of ATc (100 ng/mL) to predeplete targets. For CRISPRi strains, ATc was maintained in the media at the same concentration until cells were harvested. One OD unit was used to seed filters for growth on 7H10-ADC agar plates. After 5 days of growth, bacteria-laden filters were transferred and floated on 7H9-ADC media in a 'swimming pool' set up overnight. Filters were then transferred and floated on 7H9-OADC swimming pools and incubated for 24 hr. For metabolite extraction, filters were transferred to 1 mL of acetonitrile:methanol:water (2:2:1). Bacteria were disrupted by bead beating six times at 6000 rpm for 30 s at 4 °C (Precellys), and lysates were clarified by centrifugation and filter sterilized as described above. Lysates were transferred into mass spectrometry vials and analyzed using a semi-quantitative LC/MS-based ion pairing method. Briefly, samples were separated on a Zorbax Extend C18 column (Agilent). The mobile phase consisted of solvent A (97:3 water:methanol) and solvent B (100% methanol), both containing 5 mM tributylamine and 5.5 mM acetic acid, and the gradient used was as follows: 0–3.5 min, 0% B; 4–7.5 min, 30% B; 8–15 min, 35% B; 20–24 min, 99% B; 24–24.5 min, 0% B, followed by a 10 min re-equilibration period at 0% B at a flow rate of 0.25 mL/min. This was achieved using an Agilent 1200 Series liquid chromatography (LC) system coupled to an Agilent 6220 accurate mass time of flight (TOF) mass spectrometer in negative acquisition mode, and 5 μL of the sample were injected for each run. Dynamic mass axis calibration was achieved by continuous infusion of a reference mass solution using an isocratic pump with a 100:1 splitter. Resulting data were analyzed using Agilent MassHunter Qualitative Analysis Navigator software. Relative abundances of metabolites were determined by quantitation of peak heights or areas with a mass tolerance of +/-35 ppm. Absolute abundances of cAMP were determined by comparing sample peak heights (using $m/z$ of 328.0452 for the cAMP (M-H)$^-$ ion (RT = 9.0 min)) to a standard curve generated by spiking cAMP (final concentration range 0.015 μM to 3.9 μM) into the mycobacterial lysate. In all cases, ion counts were normalized to residual protein content in the samples, which was measured using a BCA assay (Pierce).

## Measuring camp by ELISA

Strains were grown to an OD of 0.8–1.0 in 7H9-ADC media. CRISPRi strains were grown for 5 days to an OD of 0.8–1.0 in the presence of ATc (100 ng/mL) to predeplete targets. For CRISPRi strains, ATc was maintained in the media at the same concentration until cells were harvested. 2.5 OD units were used to inoculate 5 mL cultures of 7H9-ADC or 7H9-OADC. Antibiotics were added to 7H9-OADC at a concentration of 10 nM. After 24 hr of growth, the entire culture was harvested by centrifugation (4000 × g, 10 min). Pellets were washed twice by resuspending in 800 μL cold PBS and centrifugation (15,000 × g, 5 min, 4 °C) before resuspending in 400 μL lysis buffer (50 mM Tris-Cl (pH 8.2), 100 mM NaCl, 10% (v/v) glycerol, 10 mM β-mercaptoethanol, protease inhibitor cocktail (Sigma-Aldrich, #11873580001)). Cells were lysed by three bead beating cycles at 10,000 rpm at 4 °C for 60 s (Precellys), resting on a cooled metal rack for 60 s between each cycle. Beads and debris were centrifuged (15,000 × g, 1 min, 4 °C) before transferring to a new tube for further lysate clarification by centrifugation (15,000 × g, 20 min, 4 °C). This supernatant was filter sterilized. Protein concentrations were quantified by Qubit 2.0 Fluorometer (Invitrogen). 200 μL of 0.2 M HCl was added to 200 μL of sample and heated at 95 °C for 10 min and diluted at least 1:5 for quantification by Direct cAMP ELISA kit with acetylation (Enzo #ADI-900–066 A). Resulting data were analyzed using GraphPad Prism.

## Lipid uptake assays

Vented T-25 tissue culture flasks were used to culture Mtb in 7H9 containing fatty acid-free albumin-dextrose supplemented with 0.01% glycerol and 0.05% tyloxapol. Once the bacteria reached the mid-logarithmic growth phase, cells were concentrated in a spent medium to a final $OD_{600}$ of 0.7. The bacterial cultures were immediately supplied 1.0 μCi of [1-$^{14}$C]-oleic acid (Perkin Elmer) and were incubated at 37 °C. At the 5, 30, 60, and 120 min time points 1.5 ml of the bacterial samples were removed. Each sample was washed three times with cold PBS containing 0.1% Triton X-100 and 0.1% fatty acid-free BSA to remove surface-bound radiolabel. Following the three washes the bacteria were fixed in 4% paraformaldehyde (PFA) and $^{14}$C incorporation was quantified via scintillation counting. Radioactive counts at each time point were plotted and used for linear regression calculations to

determine the rate of lipid uptake. The rates were normalized to wild type and expressed as uptake efficiency (%).

## Radiorespirometry assays

Catabolism of fatty acids was assessed by measuring the amount of $^{14}CO_2$ released from [1-$^{14}$C]-oleic acid (Perkin Elmer). Cultures of Mtb were grown in 7H9 containing fatty acid-free albumin-dextrose supplemented with 0.01% glycerol and 0.05% tyloxapol to the mid-logarithmic phase of growth in vented T-25 tissue culture flasks. Cultures were then concentrated in spent medium to an $OD_{600}$ of 0.5 and 1.0 µCi of radiolabeled lipid was added to each flask. Flasks were individually sealed in an air-tight container along with an open vial containing 0.5 ml of 1 M NaOH. After 5 hr of incubation at 37 °C, the NaOH was neutralized with 0.5 ml of 1 M HCl and the amount of $Na_2{}^{14}CO_3$ present was quantified via scintillation counting. Values were expressed as $\%CO_2$ release relative to the radioactive counts for the wild type.

## Acknowledgements

We thank Michael Berney and members of the Rock Lab for comments on the manuscript and/or helpful discussions. We thank Christopher Sassetti for the Mce1 antibodies. We thank Bob Mackie for his influence on camp art and fashion, which inspired the name 'MacE' for Rv3645. This work was supported by the Bill and Melinda Gates Foundation (INV-004709, KYR), the NIH (P01AI143575, SE, DS; R01130018, BCV), a joint NIH tuberculosis research units network (TBRU-N) grant (U19AI162584, JMR, KYR), the Department of Defense (PR192421, J.M.R.), the Robertson Therapeutic Development Fund (JMR), and an NIH/NIAID New Innovator Award (1DP2AI144850-01, JMR).

## Additional information

### Funding

| Funder | Grant reference number | Author |
| --- | --- | --- |
| Bill and Melinda Gates Foundation | INV-004709 | Kyu Y Rhee |
| National Institutes of Health | P01AI143575 | Dirk Schnappinger Sabine Ehrt |
| National Institutes of Health | R01130018 | Brian C VanderVen |
| NIH Tuberculosis Research Units Network | U19AI162584 | Kyu Y Rhee |
| Department of Defense | PR192421 | Jeremy Rock |
| Robertson Therapeutic Development Fund | | Jeremy Rock |
| NIH/NIAID New Innovator Award | 1DP2AI144850-01 | Jeremy Rock |

The funders had no role in study design, data collection and interpretation, or the decision to submit the work for publication.

### Author contributions

Andrew I Wong, Conceptualization, Resources, Data curation, Formal analysis, Validation, Investigation, Visualization, Methodology, Writing – original draft, Project administration, Writing – review and editing; Tiago Beites, Conceptualization, Resources, Formal analysis, Validation, Investigation, Methodology, Writing – original draft, Writing – review and editing; Kyle A Planck, Rachael A Fieweger, Resources, Formal analysis, Investigation, Visualization, Methodology, Writing – original draft, Writing – review and editing; Kathryn A Eckartt, Software, Formal analysis, Methodology, Writing – review and editing; Shuqi Li, Nicholas C Poulton, Investigation, Methodology, Writing – review and editing; Brian C VanderVen, Conceptualization, Formal analysis, Supervision, Funding acquisition, Investigation,

Methodology, Project administration, Writing – review and editing; Kyu Y Rhee, Conceptualization, Resources, Supervision, Funding acquisition, Methodology, Project administration, Writing – review and editing; Dirk Schnappinger, Sabine Ehrt, Conceptualization, Supervision, Funding acquisition, Methodology, Project administration, Writing – review and editing; Jeremy Rock, Conceptualization, Supervision, Funding acquisition, Methodology, Writing – original draft, Project administration, Writing – review and editing

### Author ORCIDs
Shuqi Li ⓘ http://orcid.org/0000-0002-8133-6838
Brian C VanderVen ⓘ http://orcid.org/0000-0003-3655-4390
Sabine Ehrt ⓘ http://orcid.org/0000-0002-7951-2310
Jeremy Rock ⓘ http://orcid.org/0000-0002-9310-951X

### Decision letter and Author response
Decision letter https://doi.org/10.7554/eLife.81177.sa1
Author response https://doi.org/10.7554/eLife.81177.sa2

## Additional files

### Supplementary files
• Supplementary file 1. Spreadsheet of plasmids, sgRNAs, and primers used in this work.
• MDAR checklist

### Data availability
RNA-seq data of Mtb H37Rv are deposited in NCBI's Sequence Read Archive (SRA) under BioProject PRJNA930437. Whole genome sequencing data for Δ*macE* and derived spontaneous rescue mutants were deposited in NCBI's SRA under BioProject PRJNA811534. CRISPRi suppressor screen sequencing data was deposited in NCBI's SRA under BioProject PRJNA814682.

The following datasets were generated:

| Author(s) | Year | Dataset title | Dataset URL | Database and Identifier |
|---|---|---|---|---|
| Wong AI, Rock JM | 2022 | Rv3645 palmitic acid CRISPRi suppressor screen | http://www.ncbi.nlm.nih.gov/bioproject/?term=PRJNA814682 | NCBI BioProject, PRJNA814682 |
| Beites T, Schnappinger D, Ehrt S | 2022 | Mycobacterium tuberculosis rv3645 knockout | http://www.ncbi.nlm.nih.gov/bioproject/?term=PRJNA811534 | NCBI BioProject, PRJNA811534 |
| Wong AI, Rock JM | 2022 | WT Mtb H37Rv RNA-seq | http://www.ncbi.nlm.nih.gov/bioproject/?term=PRJNA930437 | NCBI BioProject, PRJNA930437 |

The following previously published dataset was used:

| Author(s) | Year | Dataset title | Dataset URL | Database and Identifier |
|---|---|---|---|---|
| Li S, Poulton NC, DeJesus MA, Rock JM | 2021 | Large scale chemical-genetic CRISPRi screen in Mycobacterium tuberculosis | https://www.ncbi.nlm.nih.gov/bioproject/?term=PRJNA738381 | NCBI BioProject, PRJNA738381 |

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
