## [Editor Report]

Bacteria living in stressful and fluctuating environments need to respond to changing conditions, and many species, including *Mycobacterium tuberculosis*, the causative agent of tuberculosis, use cyclic AMP (cAMP) as a secondary messenger to sense and respond to specific stimuli. What distinguishes *M. tuberculosis*, is that its genome encodes at least 15 adenylate cyclases, enzymes that synthesize cAMP from ATP. Using state-of-the-art methods in this important study, the authors characterized one specific adenylate cyclase, Rv3645, and convincingly demonstrate that it is the most significant contributor to cAMP levels, in addition to mediating fatty acid metabolism and antibiotic resistance. This manuscript will be of broad interest to readers in the field of tuberculosis drug discovery and bacterial metabolism.

---

## [Decision Letter]

**Decision letter after peer review:**

Thank you for submitting your article "Cyclic AMP is a critical mediator of intrinsic drug resistance and fatty acid metabolism in *M. tuberculosis*" for consideration by *eLife*. Your article has been reviewed by 3 peer reviewers, and the evaluation has been overseen by a Reviewing Editor and Bavesh Kana as the Senior Editor. The following individual involved in the review of your submission has agreed to reveal their identity: William Jacobs (Reviewer #2).

Essential revisions:

1. The major objective of this study is to investigate the role of cAMP levels in LCFA metabolism. Thus, experiments monitoring the cAMP levels of ΔRv3645, ΔRv3645-Rv1339 KD, and WT-Rv1339 OE in media with/without LCFA with/without antibiotics are necessary to support the conclusions. Also, as shown in Supplementary Figure 1, only Rv3645 showed functional essentiality in resisting antibiotics. Thus, the authors include at least one adenylate cyclase (eg., Rv2212 as this enzyme is known to sense unsaturated fatty acids) as a negative control.

2. The reduction of cAMP levels is proposed as the primary mechanistic basis underlying ΔRv3645 hypersensitivity to multiple antibiotics. cAMP is synthesized by hydrolyzing ATP and releasing PPi. Thus, reduced biosynthesis of cAMP in Δrv3645 may be associated with the accumulation of ATP and depletion of PPi. As these metabolites are usually tightly regulated, increased drug sensitivity may be directly or indirectly associated with a dysregulation in ATP and PPi abundance, in addition to cAMP depletion. The authors should report ATP and PPi concentrations of Δrv3645 and Δrv3645-rv1339 OE.

3. Lines 182 – 184, sensitivity was not observed towards the odd chain fatty acids propionic acid and valeric acid, nor to cholesterol, ruling out propionate-derived toxicity as the source of the fatty acid sensitive growth phenotype. LCFA metabolism also can be toxic to *M. tuberculosis* if ICL aldehyde end products and ketoacidosis metabolites are accumulated, as reported in PMID: 28265055. In this context, quantification of ICL aldehyde end products and acetyl-CoA-derived ketoacidosis metabolites in ΔRv3645 and Rv3645KD is important.

4. Figure 5C, D: cAMP quantification should include the condition using media without LCFA as a negative control. The cAMP quantification experiment should also be done after treatment with antibiotics using LCFA. Figure 5G: cAMP level restoration should be confirmed using rv3645KD-rv3645cs and rv3645KD-rv3645cs-V59.

5. Reviewers did express concern regards PDIM, which can be lost during in vitro passage. Have the authors assayed their strains for the presence of PDIM? Or assessed these phenotypes in strains known to have PDIM?

Other issues that must be addressed:

1. Figure 1B: Provide the concentrations of each antibiotic used.

2. Figure 2B: Antibiotic uptake values of rv3645 KD should be included.

3. Line 211: lipid metabolism, should read "long chain fatty acid metabolism".

4. Figure 1 legend, Please describe FDR.

5. Figure 2. It is not clear from the Figure or methods, what the concentration of BODIPY-Vancomycin used (2 ug/ml) corresponds to in terms of MIC. If the rv3645 KD strain is hypersensitive to Vancomycin, there is a possibility that the drug uptake measurements are biased towards cells that are still intact and have not yet been lysed, which might be similar to wild-type MTB. Maybe the rv3645 KD cells that hyper-accumulate Vancomycin are all lysed rapidly. Have the authors measured the kill-kinetics of Vancomycin over time? Or make a comparative assessment of values reported in the literature?

6. Figure 4. Rv1339 was identified in the suppressor screen and validated in subsequent experiments. What about Rv0805, the other main PDE? Did this come up on the screen? Also looking at the data in supplementary Figure 1a, interestingly the phenotype of rv0805 is opposite to that of rv1339 and mirrors that of rv3645. How can these observations be reconciled? Perhaps the authors can discuss this.

*Reviewer #1 (Recommendations for the authors):*

The study by Wong et al. investigates the role of Rv3645, one among 15 *M. tuberculosis* adenylated cyclase enzymes, and cyclic AMP as mediators that enable *M. tuberculosis* to catalyze LCFA (long chain fatty acid) to mitigate LCFA-induced metabolic toxicity and gain accompanied intrinsic drug tolerance. To this end, this study used the genome-wide CRISPRi library screening data (a previously published work) to pinpoint the rv3645 gene as a key candidate for the *M. tuberculosis* intrinsic drug tolerance. Using CRISPRi KD and genetic KO techniques targeting rv3645 and a Δrv3645 suppressor screen technique, authors observed that Rv3645 is the main functional adenylate cyclase enzyme used to biosynthesize cAMP for an effective LCFA metabolism and antibiotic effect evasion. This study is well-designed and experimentally solid to prove a new function of Rv3645. However, there are a couple of ways to improve the mechanistic bases, as noted below.

1. The major objective of this study is to propose the role of cAMP level in LCFA metabolism in *M. tuberculosis* intrinsic drug tolerance. Thus, this reviewer recommends the experiments monitoring the cAMP levels of Δrv3645, Δrv3645-rv1339 KD, and WT-rv1339 OE in media with/without LCFA with/without antibiotics. Also, as shown in Supplementary Figure 1, only rv3645 showed functional essentiality in resisting antibiotics. Thus, the authors include at least one adenylate cyclase (eg., rv2212 as this enzyme is known to sense unsaturated fatty acids) as a negative control.

2. Authors propose that reduction of cAMP levels is a main mechanistic basis underlying Δrv3645 hypersensitivity to multiple antibiotics. As already known, cAMP is synthesized by hydrolyzing ATP and releasing PPi. Thus, reduced biosynthesis of cAMP in Δrv3645 may be associated with the accumulation of ATP and depletion of PPi. As ATP and PPi concentrations are tightly regulated in gaining drug tolerance of *M. tuberculosis*, Δrv3645 drug sensitivity may be directly or indirectly associated with a dysregulation in ATP and PPi abundance in addition to cAMP depletion. Thus, to prove that Δrv3645 hypersensitivity to antibiotic effects is mainly due to cAMP depletion, this reviewer recommends including ATP and PPi concentrations of Δrv3645 and Δrv3645-rv1339 OE.

3. Lines 182 – 184 sensitivity was not observed towards the odd chain fatty acids propionic acid and valeric acid nor to cholesterol, ruling out propionate-derived toxicity as the source of the fatty acid sensitive growth phenotype: Even LCFA metabolism also can be toxic to *M. tuberculosis* if its ICL aldehyde endproducts and ketoacidosis metabolites are highly accumulated as reported in PMID: 28265055. Thus, this reviewer recommends the quantification of ICL aldehyde endproducts and acetyl-CoA-derived ketoacidosis metabolites in Δrv3645 and rv3645KD.

*Reviewer #2 (Recommendations for the authors):*

One possible weakness in this work is that the phenotypes observed are strain specific. Did the authors perform similar experiments on other strains of TB?

*Reviewer #3 (Recommendations for the authors):*

– Figure 1 legend, Please describe FDR.

– Figure 2. It is not clear from the Figure or methods, what the concentration of BODIPY-Vancomycin used (2 ug/ml) corresponds to in terms of MIC. If the rv3645 KD strain is hypersensitive to Vancomycin, there is a possibility that the drug uptake measurements are biased towards cells that are still intact and have not yet been lysed, which might be similar to wild-type MTB. Maybe the rv3645 KD cells that hyper-accumulate Vancomycin are all lysed rapidly. Have the authors measured the kill-kinetics of Vancomycin over time?

– Figure 4. Rv1339 was identified in the suppressor screen and validated in subsequent experiments. What about Rv0805, the other main PDE? Did this come up in the screen? Also looking at the data in supplementary Figure 1a, interestingly the phenotype of rv0805 is opposite to that of rv1339 and mirrors that of rv3645. How can these observations be reconciled? Maybe the authors can discuss this.

– Figure 5H. The authors have speculated on the likely link between fatty acid metabolism and antibiotic sensitivity through activation of Mt-Pat and redox imbalance. These are easily testable hypotheses through expression analysis or metabolite analysis. This would greatly strengthen and fully validate their hypothesis.

---

## [Author Response]

Essential revisions:1. The major objective of this study is to investigate the role of cAMP levels in LCFA metabolism. Thus, experiments monitoring the cAMP levels of ΔRv3645, ΔRv3645-Rv1339 KD, and WT-Rv1339 OE in media with/without LCFA with/without antibiotics are necessary to support the conclusions. Also, as shown in Supplementary Figure 1, only Rv3645 showed functional essentiality in resisting antibiotics. Thus, the authors include at least one adenylate cyclase (eg., Rv2212 as this enzyme is known to sense unsaturated fatty acids) as a negative control.

To address the reviewers’ comment, we quantified cAMP levels in the following Mtb strains: WT, ∆*rv3645*+EV, ∆*rv3645+rv3645,* ∆*rv3645*+*rv1339* KD, and WT+*rv1339* OE. Strains were grown for 24 hours with/without oleic acid and with/without antibiotics. As can be seen in Figure 5 —figure supplement 5A, Author response image 1 and Author response image 2 the presence of oleic acid in the growth medium does not dramatically alter cAMP levels. Similar results were obtained when quantifying cAMP by ELISA or by mass spectrometry. We note that the magnitude of cAMP detected by ELISA is lower than quantified by mass spectrometry, likely reflecting differences in culture conditions and lysate preparation, but the relative changes in cAMP levels measured between strains remains similar across experiments.

**Author response image 1. sa2fig1:** 

The complemented strain ∆*rv3645+rv3645* produces higher cAMP levels under most conditions tested, consistent with the complementation construct modestly overexpressing *rv3645* from the *hsp60* promoter, as determined by RT-qPCR. To further address how cAMP may regulate LCFA catabolism in Mtb, we collaborated with the VanderVen lab to quantify [1-^14^C]-oleic acid uptake and metabolism in the following Mtb strains: WT, ∆*rv3645*+EV, ∆*rv3645+rv3645,* WT+*rv1339* OE and WT+*rv1339*(*D180A*) OE*.* As can be seen in Author response image 3 and Figure 5—figure supplement 3, reduced cAMP levels correlate with elevated oleic acid uptake and metabolism. Alterations in oleic acid uptake and metabolism were not mediated by altered levels of Mce1A, Mce1D, Mce1E, nor MceG levels. Our observation of elevated oleic acid uptake at low cAMP levels is consistent with prior results from the VanderVen lab that observed reduced oleic acid uptake in *rv1339::Tn* Mtb mutants in infected macrophages (PMID: 30735132). These results suggest that loss of *rv1339* suppresses the LCFA toxicity in ∆*rv3645* by reducing LCFA catabolism.

**Author response image 3. sa2fig3:** 

To test the effect of antibiotic treatment on cAMP levels, we exposed the same set of strains to 10 μM vancomycin (VAN), rifampicin (RIF), or linezolid (LZD) for 24 hours. All drugs were tested above their MIC90. cAMP was measured by ELISA. Whereas vancomycin did not alter cAMP levels, both rifampicin and linezolid led to large increases in cAMP in a manner partially dependent on *rv3645*. As shown in Figure 1D-G of the manuscript, *rv3645* KD are more sensitive to vancomycin and rifampicin but not linezolid. Thus, there is no correlation between the drugs that induce cAMP levels and those drugs that *rv3645* KD is sensitized to. We note that the ∆*rv3645*+*rv1339* KD strain shows only a modest increase in cAMP following LZD treatment in this particular experiment, but replicate experiments showed stronger cAMP induction in this strain, comparable to that observed in the WT strain. Taken together, these results suggest that cAMP produced by Rv3645 is necessary for H37Rv Mtb to maintain homeostasis, both with respect to LCFA uptake and catabolism and intrinsic drug resistance. While H37Rv Mtb can respond to alterations in cellular physiology, e.g. inhibition of transcription and/or translation, by altering cAMP levels in a manner largely dependent on Rv3645, it does not appear that an inducible cAMP stress response is critical to modulate LCFA uptake and catabolism nor intrinsic drug resistance.

To address the reviewers’ comment regarding the functional essentiality of Rv3645 in resisting antibiotics, we further validated the screen results by showing that knockdown of the alternative adenylate cyclases *rv2212*, *rv1264*, or *rv1625c* does not alter sensitivity to vancomycin.

These data are now included as Figure 3E-F, Figure 5F-G, and Figure 5 – figure supplements 3 and 5 in the revised manuscript and Author response image 4.

**Author response image 4. sa2fig4:** 

2. The reduction of cAMP levels is proposed as the primary mechanistic basis underlying ΔRv3645 hypersensitivity to multiple antibiotics. cAMP is synthesized by hydrolyzing ATP and releasing PPi. Thus, reduced biosynthesis of cAMP in Δrv3645 may be associated with the accumulation of ATP and depletion of PPi. As these metabolites are usually tightly regulated, increased drug sensitivity may be directly or indirectly associated with a dysregulation in ATP and PPi abundance, in addition to cAMP depletion. The authors should report ATP and PPi concentrations of Δrv3645 and Δrv3645-rv1339 OE.

We thank the reviewers for this suggestion. To address the reviewers’ comment, we quantified ATP and PPi levels by mass spectrometry in the following Mtb strains: WT, ∆*rv3645*+EV, ∆*rv3645+rv3645,* ∆*rv3645*+*rv1339* KD, and WT+*rv1339* OE. Strains were grown for 24 hours with a growth-inhibitory concentration of oleic acid. As can be seen in Figure 5 —figure supplement 2, ~10-30-fold reductions in cAMP levels (Figure 5C-E) do not dramatically alter ATP nor PPi levels in these strains.

3. Lines 182 – 184, sensitivity was not observed towards the odd chain fatty acids propionic acid and valeric acid, nor to cholesterol, ruling out propionate-derived toxicity as the source of the fatty acid sensitive growth phenotype. LCFA metabolism also can be toxic to M. tuberculosis if ICL aldehyde end products and ketoacidosis metabolites are accumulated, as reported in PMID: 28265055. In this context, quantification of ICL aldehyde end products and acetyl-CoA-derived ketoacidosis metabolites in ΔRv3645 and Rv3645KD is important.

This is a great suggestion and a hypothesis that we entertained early in this project. As the reviewer points out, LCFA can be toxic if isocitrate lyase (ICL) aldehyde end products accumulate. As reported in PMID: 28265055, ICL aldehyde end products accumulate when Mtb is deficient in malate synthase, encoded by the *glcB* gene. We reasoned that if ICL aldehyde end product accumulation was a source of LCFA toxicity and drug sensitivity in ∆*rv3645*, then ∆*glcB* and ∆*rv3645* Mtb strains should share similar phenotypes. This was generally not the case , and thus we concluded that the phenotypes observed in ∆*rv3645* Mtb strains occur by a mechanism distinct from ICL aldehyde end product accumulation.

Fatty acid toxicity: similar to ∆*rv3645*, ∆*glcB* mutants can be recovered in media lacking fatty acids, and fatty acids are toxic even in the presence of the alternative carbon sources glycerol and glucose (PMID: 28265055). However, the fatty acid susceptibility profiles of ∆*rv3645* and ∆*glcB* are distinct. Whereas ∆*rv3645* is only susceptible to LCFA including palmitic acid, oleic acid, and arachidonic acid (Figure 3B-D, Figure 3 – figure supplement 1B-G), ∆*glcB* is sensitive to short, medium, and long-chain fatty acids including acetate, propionate, butyrate, valerate, palmitic acid, oleic acid, and cholesterol (PMID: 28265055). Moreover, whereas chemical inactivation of Icl1 rescued growth of ∆*glcB* in the presence of LCFA (PMID: 28265055), we did not observe that knockdown of *icl1* rescued the palmitic acid toxicity in ∆*rv3645* in the suppressor screen (Figure 4A-B).

Intrinsic drug resistance: the chemical-genetic interactions observed for *rv3645* KD and *glcB* KD in our published CRISPRi screens are distinct (see Author response image 5, PMID: 35637331).

**Author response image 5. sa2fig5:** 

Finally, to directly test if ICL aldehyde end product accumulation was a source of toxicity in ∆*rv3645* strains, we quantified the abundance of glyoxylate, butyryl-coenzyme A (butyryl-CoA), acetoacetate, and β-hydroxybutyrate by mass spectrometry in the following Mtb strains grown in 7H9-OADC: WT, *rv3645* KD + *rv3645*^CS^, *rv3645* KD + *rv3645*^CR^, and ∆*rv3645*. Consistent with our hypothesis that ICL aldehyde end product accumulation is not the source of fatty acid toxicity in ∆*rv3645*, the levels of these metabolites were unchanged across the tested Mtb strains. See Author response image 6.

**Author response image 6. sa2fig6:** 

4. Figure 5C, D: cAMP quantification should include the condition using media without LCFA as a negative control. The cAMP quantification experiment should also be done after treatment with antibiotics using LCFA. Figure 5G: cAMP level restoration should be confirmed using rv3645KD-rv3645cs and rv3645KD-rv3645cs-V59.

Please see our response to Essential Revision 1 to address the comments concerning Figure 5C-D.

Regarding the reviewer’s comment about Figure 5G, we first confirmed that 10 µM V-59 rescues the vancomycin sensitivity of ∆*rv3645* (the figure in the original manuscript submission utilized CRISPRi knockdown strains).

Next, we quantified cAMP by ELISA in the following Mtb strains: WT, ∆*rv3645,* and ∆*rv3645 + rv3645* in the presence or absence of 10 µM V-59. We observe higher levels of cAMP in all V-59 treated cultures, presumably explaining the rescue of ∆*rv3645* vancomycin sensitivity in the presence of V-59.

We replace Figure 5G with dose response curves of the ∆*rv3645* strains and include the cAMP measurements as Figure 5 – supplement 4.

5. Reviewers did express concern regards PDIM, which can be lost during in vitro passage. Have the authors assayed their strains for the presence of PDIM? Or assessed these phenotypes in strains known to have PDIM?

As the reviewers note, PDIM is easily lost during in vitro passaging of Mtb. The Mtb strains used in this manuscript have undergone several passages to sequentially clone CRISPRi and complementation/expression plasmids, or to engineer gene deletions, and thus we would not be surprised if our strains are PDIM–. On-going efforts in the lab to engineer uniformly PDIM+ genetic mutants in Mtb H37Rv have been unsuccessful. Thus, until such time a method exists to reliably maintain PDIM production in in vitro passaged Mtb, there is unfortunately not much that can be done to reliably assess the role of PDIM in these phenotypes.

Other issues that must be addressed:1. Figure 1B: Provide the concentrations of each antibiotic used.

Corrected. These data are now included as Figure 1 —figure supplement 2 in the revised manuscript.

2. Figure 2B: Antibiotic uptake values of rv3645 KD should be included.

Corrected. These data are now included as Figure 2 —figure supplement 2 in the revised manuscript.

3. Line 211: lipid metabolism, should read "long chain fatty acid metabolism".

Corrected.

4. Figure 1 legend, Please describe FDR.

Corrected.

5. Figure 2. It is not clear from the Figure or methods, what the concentration of BODIPY-Vancomycin used (2 ug/ml) corresponds to in terms of MIC. If the rv3645 KD strain is hypersensitive to Vancomycin, there is a possibility that the drug uptake measurements are biased towards cells that are still intact and have not yet been lysed, which might be similar to wild-type MTB. Maybe the rv3645 KD cells that hyper-accumulate Vancomycin are all lysed rapidly. Have the authors measured the kill-kinetics of Vancomycin over time? Or make a comparative assessment of values reported in the literature?

The concentration of BODIPY-vancomycin used (2 µg/mL or 1.16 µM) is above the MIC90 of the *rv3645* KD strain but below the MIC90 of the NT sgRNA and complemented control strains. We do not believe this should produce artifactual results because the BODIPY-vancomycin binding assays were performed over a short time period: 30 minutes or ~2% of the total time of a single Mtb cell cycle. We first confirmed that BODIPY-vancomycin showed similar potency to vancomycin. Because BODIPY-vancomycin is much more expensive than vancomycin, we next performed time-kill experiments using 1.16 µM vancomycin with the following Mtb strains: NT sgRNA, *rv3645* KD + *rv3645^CS^*, and *rv3645* KD + *rv3645^CR^*. 30 minutes of vancomycin exposure did not cause a drop in viable CFU for any of the strains. Thus, we do not think that the results of the BODIPY-vancomycin uptake experiment are influenced by the presence of dead or dying Mtb.

These data have been included as Figure 2 —figure supplement 1.

6. Figure 4. Rv1339 was identified in the suppressor screen and validated in subsequent experiments. What about Rv0805, the other main PDE? Did this come up on the screen? Also looking at the data in supplementary Figure 1a, interestingly the phenotype of rv0805 is opposite to that of rv1339 and mirrors that of rv3645. How can these observations be reconciled? Perhaps the authors can discuss this.

Existing evidence suggests that Rv1339 is the primary 3’,5’-cAMP PDE in Mtb, as detailed below.

We did not identify *rv0805* as a suppressor of palmitic acid toxicity in the ∆*rv3645* suppressor screen. Although Rv0805 is a cAMP phosphodiesterase, it is 150 times more active against 2’,3’-cAMP (a cAMP species associated with RNA degradation) than 3’,5’-cAMP (the product of adenylate cyclase) (PMID: 18757371). Consistent with this poor activity towards 3’,5’-cAMP, overexpression of Rv0805 resulted in only modest (~40%) reduction in 3’,5’-cAMP in Mtb (PMID: 19516256). Lastly, *rv0805* has a restricted phylogenetic distribution, being found in Mtb and closely related bacterial species, but not in other mycobacteria like *M. smegmatis* that also encode numerous adenylate cyclases (PMID: 35718063). Indeed, it is this restricted phylogenetic distribution of *rv0805* and the fact that *M. smegmatis* protein lysate was known to harbor 3’,5’-cAMP phosphodiesterase activity that was the rationale that ultimately led to the discovery of Rv1339 as an atypical 3’,5’-cAMP phosphodiesterase (PMID: 35718063). Thus, it appears the major role of Rv0805 in mycobacterial physiology is independent of 3’,5’-cAMP. We have included this section in a revised Discussion section of the manuscript.

Reviewer #2 (Recommendations for the authors):One possible weakness in this work is that the phenotypes observed are strain specific. Did the authors perform similar experiments on other strains of TB?

We thank Bill for this astute observation. In this work we focused solely on the reference Mtb strain H37Rv. Prior CRISPRi screening results suggest that *rv3645* may be a differentially essential gene between different Mtb strains (PMID: 34297925). To test the importance of cAMP in other strains of Mtb, we cloned the *rv3645* KD and *rv1339* OE plasmids into the lineage 2 strain HN878, in which *rv3645* appears non-essential (PMID: 34297925), and tested the resulting strains in dose-response MIC assays and agar plate-based fatty acid sensitivity spotting assays. We find that knockdown of *rv3645* or overexpression of *rv1339* in HN878 results in an intermediate sensitization to vancomycin compared to H37Rv. Moreover, overexpression of *rv1339* sensitized HN878 to palmitic acid. These results suggest a role for cAMP in mediating intrinsic drug resistance and fatty acid metabolism in Mtb beyond the reference strain H37Rv.

These data have been included as Figure 5 —figure supplement 6.

Reviewer #3 (Recommendations for the authors):Specific comments:– Figure 1 legend, Please describe FDR.– Figure 2. It is not clear from the Figure or methods, what the concentration of BODIPY-Vancomycin used (2 ug/ml) corresponds to in terms of MIC. If the rv3645 KD strain is hypersensitive to Vancomycin, there is a possibility that the drug uptake measurements are biased towards cells that are still intact and have not yet been lysed, which might be similar to wild-type MTB. Maybe the rv3645 KD cells that hyper-accumulate Vancomycin are all lysed rapidly. Have the authors measured the kill-kinetics of Vancomycin over time?– Figure 4. Rv1339 was identified in the suppressor screen and validated in subsequent experiments. What about Rv0805, the other main PDE? Did this come up in the screen? Also looking at the data in supplementary Figure 1a, interestingly the phenotype of rv0805 is opposite to that of rv1339 and mirrors that of rv3645. How can these observations be reconciled? Maybe the authors can discuss this.– Figure 5H. The authors have speculated on the likely link between fatty acid metabolism and antibiotic sensitivity through activation of Mt-Pat and redox imbalance. These are easily testable hypotheses through expression analysis or metabolite analysis. This would greatly strengthen and fully validate their hypothesis.

We thank the reviewer for this comment. We agree that further analysis of phenotypes regulated by cAMP would strengthen our hypotheses. To this end, we have begun to interrogate the connection of cAMP with fatty acid metabolism. Please see our response to Essential Revision 1, demonstrating strains with *rv3645* deleted or overexpressing *rv1339* show increased uptake and catabolism of ^14^C-oleic acid.